# A Note on Sparse Generalized Eigenvalue Problem

**Yunfeng Cai, Guanhua Fang, Ping Li**
Cognitive Computing Lab
Baidu Research
No. 10 Xibeiwang East Road, Beijing 100193, China
10900 NE 8th St. Bellevue, Washington 98004, USA
{yunfengcai, guanhuafang, liping11}@baidu.com

## Abstract

The sparse generalized eigenvalue problem (SGEP) aims to find the leading eigenvector with sparsity structure. SGEP plays an important role in statistical learning and has wide applications including, but not limited to, sparse principal component analysis, sparse canonical correlation analysis and sparse Fisher discriminant analysis, etc. Due to the sparsity constraint, the solution of SGEP entails interesting properties from both numerical and statistical perspectives. In this paper, we provide a detailed sensitivity analysis for SGEP and establish the rate-optimal perturbation bound under the sparse setting. Specifically, we show that the bound is related to the perturbation/noise level and the recovery of the true support of the leading eigenvector as well. We also investigate the estimator of SGEP via imposing a non-convex regularization. Such estimator can achieve the optimal error rate and can recover the sparsity structure as well. Extensive numerical experiments corroborate our theoretical findings via using alternating direction method of multipliers (ADMM)-based computational method.

## 1   Introduction

The sparse generalized eigenvalue problem (SGEP) is to solve the following constrained optimization problem:

$$\max_{\mathbf{x} \in \mathbb{R}^p} \frac{\mathbf{x}^{\mathrm{T}} \widetilde{A} \mathbf{x}}{\mathbf{x}^{\mathrm{T}} \widetilde{B} \mathbf{x}}, \quad \text{subject to} \quad \|\mathbf{x}\|_0 \le k, \tag{1}$$

where $\widetilde{A}$, $\widetilde{B}$ are both $p$-by-$p$ symmetric matrices and $\widetilde{B}$ is (semi) positive definite; $\|\mathbf{x}\|_0$ denotes the $\ell_0$-norm of $\mathbf{x}$, i.e., the number of nonzero entries of $\mathbf{x}$; $k$ is an integer which may be much smaller than $p$. For a fixed $k$, we let $\mathbf{x}_*$ be the solution to (1). Moreover, matrices $\widetilde{A}$ and $\widetilde{B}$ usually have the following decomposition:

$$\widetilde{A} = A + E, \qquad \widetilde{B} = B + F, \tag{2}$$

where $A$, $B$ are underlying symmetric matrices and $B$ is positive definite. However, $A$ and $B$ are unobserved in practice and they can be contaminated by some noise/perturbation matrices $E$ and $F$, respectively. Let $\lambda_1$ be the largest generalized eigenvalue of matrix pair $(A, B)$ and $\mathbf{u}_1$ be its corresponding eigenvector which is also known as the leading eigenvector. In addition, $\mathbf{u}_1$ is always assumed to be sparse, i.e., $\|\mathbf{u}_1\|_0 \ll p$. Therefore, the SGEP is essentially to find an approximation of $\mathbf{u}_1$ based on observation $(\widetilde{A}, \widetilde{B})$.

The SGEP is challenging due to the following aspects. First, the sparsity constraint optimization problem (1) is NP-hard, since it is essentially a subset selection problem [19, 20]. To recover the underlying sparsity structure, we need to try different $k$ which is computationally expensive.

35th Conference on Neural Information Processing Systems (NeurIPS 2021).

Second, due to insufficient number of samples, $\widetilde{A}$ and $\widetilde{B}$ can be both ill-conditioned or even share an approximate common null space, and perturbations $E$ and $F$ can be large. As a result, the leading eigenvector of pair $(\widetilde{A}, \widetilde{B})$ may not be a good approximation of $\mathbf{u}_1$.

Though challenging, especially when there is a limited number of data samples, SGEP and related problems have received more and more attentions in the past decade and various numerical methods are proposed to solve the SGEP. For example, in [24], the SGEP is framed as a d.c. (difference of convex functions) program and is solved as a sequence of convex programs by invoking the majorization-minimization method. In [23], the $\ell_0$ norm constraint is approximated by a continuous surrogate function, an algorithm inspired by the majorization-minimization method is developed via iteratively majorizing the surrogate function by a quadratic separable function, and a systematic way based on smoothing is proposed to deal with the singularity issue. [9] proposes a semidefinite programming method for sparse canonical correlation analysis which is one of important SGEP applications. In [30], a two-stage computational framework is proposed to solve the SGEP, in which the first stage computes an initial guess via convex relaxation, and the second stage solves a nonconvex optimization via a fixed step size steepest ascent method, and followed by the simple truncation. In [22], a general framework called sparse estimation with linear programming is proposed for sparse canonical correlation analysis. Combining the idea of $\ell_1$-penalized adaptive normalized quasi-Newton algorithm with nested orthogonal complement structure, an algorithm is proposed in [32] to solve the SGEP. [13] solves the SGEP by modifying the standard generalized orthogonal iteration with a sparsity-inducing penalty for the eigenvectors. For more comprehensive review of computations in the SGEP, please see [3, 42] and references therein.

Despite the efforts mentioned above, there is very little work investigating the optimal estimation error and no work provides theoretical guarantees whether the underlying sparsity structure of the leading eigenvector of SGEP can be recovered. In this paper, we bridge the theoretical gap in the literature. First, we provide both upper and lower bounds of the approximation error given noisy observations in the sparse generalized eigenvalue problem setting. To be specific, we prove that the difference between $\mathbf{u}_1$ and $\mathbf{x}_*$ is bounded by two parts, the perturbation error and error of not recovering the true support of $\mathbf{u}_1$. Furthermore, we show that the error bound we obtained is rate optimal. That is, the lower bound matches the upper bound up to a multiplicative constant. Second, we consider a family of non-convex penalty functions and reformulate (1) into a regularization problem. We show that such regularized estimator enjoys the merits of sparsity and nearly unbiasedness. With these nice properties, the estimator can be shown to achieve the optimal estimation error rate and can recover the true support of leading eigenvector as well. In addition, we also present a new computational algorithm. The estimation procedure adopts the Alternating Direction Method of Multipliers (ADMM, [37, 21]) and can recover the sparsity structure well under mild conditions. Local convergence theory is established for the proposed method.

The rest of paper is organized as follows. In Section 2, we introduce the sparse generalized eigenvalue problem with its applications and technical tools. In Section 3, we provide a sensitivity analysis for SGEP and establish its upper and lower perturbation bounds. In Section 4, we propose a regularized estimator via using non-convex penalization method and established the estimation bound. We also propose a computational method, non-convex SGEP algorithm (NC-SGEP), to tackle with the estimation issues. Multiple numerical results are given in Section 5 and corroborate our theory. The concluding remark is given in Section 6.

**Notations:** The calligraphic letters $\mathcal{I}, \mathcal{J}, \mathcal{K}$, and $\mathcal{S}$ are usually used to denote index sets. $|\mathcal{I}|$ denotes the cardinality of $\mathcal{I}$, e.g., $\mathcal{I} = \{i_1, i_2, \ldots, i_s\}$, where $i_1, i_2, \ldots, i_s$ are distinct integers, then $|\mathcal{I}| = s$. We use $\mathbf{x}$ to denote a generic vector in $\mathbb{R}^p$ and $\mathbf{x}^{\mathrm{T}}$ to denote its transpose. $\mathbf{x}[j]$ is the $j$th element in $\mathbf{x}$. $\mathbf{x}[\mathcal{I}]$ stands for the subvector of $\mathbf{x}$ with indices in set $\mathcal{I}$. We use $A$ to denote a matrix in $\mathbb{R}^{p \times p}$. $A[i, j]$ is the entry in the $i$th row and the $j$th column. $A[\mathcal{I}, \mathcal{J}]$ stands for the submatrix of $A$ with row indices in set $\mathcal{I}$ and column indices in set $\mathcal{J}$. If $\mathcal{I} = \mathcal{J}$, $A[\mathcal{I}, \mathcal{I}]$ is also denoted by $A_{\mathcal{I}}$. We also let $(\mathcal{I}, \mathcal{J})$ be the union of sets $\mathcal{I}$ and $\mathcal{J}$. For $\mathbf{x} \in \mathbb{R}^p$, $\mathrm{supp}(\mathbf{x})$ denotes the index set of all nonzero entries of $\mathbf{x}$. For matrix $A$, $\mathrm{supp}(A)$ denotes the index set of all nonzero rows of $A$. $I_p$ is the $p \times p$ identity matrix. $\mathrm{diag}(d_1, \ldots, d_p)$ represents a diagonal matrix with $d_1, \ldots, d_p$ being its diagonal elements. $\|\mathbf{x}\|_0$ represents the number of non-zero entries in $\mathbf{x}$; $\|\mathbf{x}\|_2 := \sqrt{\sum_j \mathbf{x}^2[j]}$; $\|A\|_2$ denotes the largest singular value of $A$; $\|A\|_F$ and $\|A\|_*$ is the Frobenius norm and nuclear norm of matrix $A$ respectively. $\|A\|_\infty := \max_{i,j} |A[i, j]|$. We write $a \gg b$ $(a \ll b)$ if $a \geq Kb$ $(a \leq b/K)$ for some sufficiently large number $K$.

## 2 Preliminary

**Applications** Sparse generalized eigenvalue problem has a wide application. Many high-dimensional multivariate statistical problems can be formulated as special instances of (1).

**Example 1** [***Sparse Principal Component Analysis** [44, 5, 1, 35, 36]]* Given $n$ observations with $p$ features, sparse principal component analysis (SPCA) seeks the best low dimensional projection of the observed data for increasing interpretability, minimizing information loss and achieving sparsity structure. Let $\widetilde{\Sigma}$ be the empirical covariance matrix. SPCA aims to solve

$$\max_{\mathbf{x} \in \mathbb{R}^p} \mathbf{x}^{\mathrm{T}} \widetilde{\Sigma} \mathbf{x}, \ \ subject \ to \ \|\mathbf{x}\|_0 \leq k, \|\mathbf{x}\|_2 = 1. \tag{3}$$

*Such problem is a special case of SGEP where $\widetilde{A} = \widetilde{\Sigma}$ and $\widetilde{B} = I$.*

**Example 2** [***Sparse Fisher's Discriminant Analysis** [31, 12, 17, 14, 16]]* Given $n$ observations with $p$ features from $K$ different classes, Fisher's discriminant analysis seeks a projection for mapping observations to the low dimensional space on which the between-class variance $\Sigma_b$ is large while the within-class variance $\Sigma_w$ is small. Let $\widetilde{\Sigma}_b$ and $\widetilde{\Sigma}_w$ be the sample estimates of $\Sigma_b$ and $\Sigma_w$, respectively. To obtain the sparse leading discriminant vector, one can solve

$$\max_{\mathbf{x} \in \mathbb{R}^p} \mathbf{x}^{\mathrm{T}} \widetilde{\Sigma}_b \mathbf{x}, \ \ subject \ to \ \mathbf{x}^{\mathrm{T}} \widetilde{\Sigma}_w \mathbf{x} = 1, \ \|\mathbf{x}\|_0 \leq k. \tag{4}$$

*Such a problem can be formulated as an SGEP with $\widetilde{A} = \widetilde{\Sigma}_b$ and $\widetilde{B} = \widetilde{\Sigma}_w$.*

**Example 3** [***Sparse Canonical Correlation Analysis** [39, 9, 10, 41]]* Given two random vectors $X \in \mathbb{R}^p$, $Y \in \mathbb{R}^p$, let $\Sigma_{xx}$, $\Sigma_{yy}$, $\Sigma_{xy}$ be the covariance matrices for $X$, $Y$, and the cross-covariance matrix between $X$ and $Y$, respectively. Let $\widetilde{\Sigma}_{xx}$, $\widetilde{\Sigma}_{yy}$, $\widetilde{\Sigma}_{xy}$ be estimators (constructed from samples) for $\Sigma_{xx}$, $\Sigma_{yy}$, $\Sigma_{xy}$, respectively. Sparse canonical correlation analysis (SCCA) aims to solve the constrained optimization problem:*

$$\max_{\mathbf{u}_x, \mathbf{u}_y} \mathbf{u}_x^{\mathrm{T}} \widetilde{\Sigma}_{xy} \mathbf{u}_y, \ \ subject \ to \ \mathbf{u}_x^{\mathrm{T}} \widetilde{\Sigma}_{xx} \mathbf{u}_x = 1, \ \mathbf{u}_y^{\mathrm{T}} \widetilde{\Sigma}_{yy} \mathbf{u}_y = 1,$$

$$\|\mathbf{u}_x\|_0 \leq k_x, \ \|\mathbf{u}_y\|_0 \leq k_y,$$

*where $k_x$ and $k_y$ are two small integers. Such a problem can be recast as an SGEP with*

$$\widetilde{A} = \begin{bmatrix} 0 & \widetilde{\Sigma}_{xy} \\ \widetilde{\Sigma}_{xy}^{\mathrm{T}} & 0 \end{bmatrix}, \quad \widetilde{B} = \begin{bmatrix} \widetilde{\Sigma}_{xx} & 0 \\ 0 & \widetilde{\Sigma}_{yy} \end{bmatrix}, \quad \mathbf{u}_1 = \begin{bmatrix} \mathbf{u}_x \\ \mathbf{u}_y \end{bmatrix}.[1]$$

*Strictly speaking, two problems are not exactly equivalent, because $\|\mathbf{u}\|_0 \leq k_x + k_y$ does not necessarily imply $\|\mathbf{u}_x\|_0 \leq k_x$ and $\|\mathbf{u}_y\|_0 \leq k_y$.*

**Technical preparation** We first introduce some useful terminologies for describing the generalized eigenvalue problem (GEP). Let $A \in \mathbb{R}^{p \times p}$ be a symmetric matrix and $B \in \mathbb{R}^{p \times p}$ be a symmetric and positive definite matrix. Then GEP for the matrix pair $(A, B)$ is defined as

$$A\mathbf{u}_i = \lambda_i B \mathbf{u}_i; \ \ i = 1, \dots, p.$$

Without loss of generality, we can always assume that $\lambda_1 \geq \cdots \geq \lambda_p$. Here $\mathbf{u}_i$ is $i$th eigenvector and $\lambda_i$ is $i$th eigenvalue; $(\lambda_i, \mathbf{u}_i)$ is called $i$th eigenpair of $(A, B)$. Notice that $\mathbf{u}_i$ is only determined up to a scale. In this paper, we always assume that $\|\mathbf{u}_i\|_2 = 1$. The perturbation analysis of generalized eigenvalue problem has been extensively studied since 1970s and many perturbation bounds have been developed [25, 27, 28, 33]. For reader convenience, we present several existing fundamental results before moving to our main theory immediately.

**Definition 1** *The angle between $\mathbf{x}, \mathbf{y} (\neq \mathbf{0}) \in \mathbb{R}^p$ is defined as $\theta(\mathbf{x}, \mathbf{y}) := \arccos \frac{|\mathbf{x}^{\mathrm{T}} \mathbf{y}|}{\|\mathbf{x}\|_2 \|\mathbf{y}\|_2}$.*

We assume $\mathbf{x}$ and $\mathbf{y} \in \mathbb{R}^p$ have unit $l_2$ norms and let $[\mathbf{x}, X_2]$ be an orthogonal matrix. Then it holds

$$|\sin \theta(\mathbf{x}, \mathbf{y})| = \|X_2^{\mathrm{T}} \mathbf{y}\|_2. \tag{5}$$

Furthermore, it can be checked that $|\sin \theta(\mathbf{x}, \mathbf{y})| \leq \|\mathbf{x} - \mathbf{y}\|_2 \leq \sqrt{2}|\sin \theta(\mathbf{x}, \mathbf{y})|$. Thus $|\sin \theta(\mathbf{x}, \mathbf{y})|$ is a measure to quantify the distance between two vectors. In generalized eigenvalue problem, Crawford number is an important quantity for characterizing the perturbation bound.

**Definition 2** *The* Crawford number *of a symmetric matrix pair* $(A, B)$ *is defined as*

$$c(A, B) := \min_{\|\mathbf{x}\|_2 = 1} \sqrt{(\mathbf{x}^T A \mathbf{x})^2 + (\mathbf{x}^T B \mathbf{x})^2}.$$

*A symmetric matrix pair* $(A, B)$ *is referred to as* "definite" *if* $c(A, B) > 0$.

Obviously, the symmetric matrix pair $(A, B)$ is definite for any positive definite $B$. By [25], it is known that the Crawford number is continuous with respect to the matrix pair.

## 3 Understanding of Perturbation Bound

### 3.1 On the General Setting

In order to understand the difference between the solution of (1) and the true leading eigenvector, we first describe the perturbation results without considering the sparsity. The following lemma is from [29] and is a generalization of the Davis-Kahn's $\sin \Theta$ theorem in [6] (also see [26, 15]). It gives an upper bound for $|\sin \theta(\mathbf{u}_1, \widetilde{\mathbf{u}}_1)|$, where $\widetilde{\mathbf{u}}_1$ is the leading eigenvector of $(\tilde{A}, \tilde{B})$.

**Lemma 1** *Suppose* $(A, B)$, $(\widetilde{A}, \widetilde{B}) = (A + E, B + F)$ *are both symmetric-definite pairs, and their eigenvalues satisfy* $\lambda_1 \geq \cdots \geq \lambda_p \geq 0$ *and* $\widetilde{\lambda}_1 \geq \cdots \geq \widetilde{\lambda}_p \geq 0$, *respectively. Let* $\phi_1 = \arctan \lambda_1 > \widetilde{\phi}_2 = \arctan \widetilde{\lambda}_2$, $\mathbf{u}_1$, $\widetilde{\mathbf{u}}_1$ *be the eigenvectors corresponding to* $\lambda_1$, $\widetilde{\lambda}_1$, *respectively. Denote*

$$C_u = \frac{\sqrt{2(\|A\|_2^2 + \|B\|_2^2)}}{c(A, B)}, \quad \xi = \frac{\sqrt{\|E \mathbf{u}_1\|_2^2 + \|F \mathbf{u}_1\|_2^2}}{c(\widetilde{A}, \widetilde{B})},$$

*then*

$$|\sin \theta(\mathbf{u}_1, \widetilde{\mathbf{u}}_1)| \leq \frac{C_u \, \xi}{\sin(\phi_1 - \widetilde{\phi}_2)}.$$

Therefore, it holds asymptotically that

$$|\sin \theta(\mathbf{u}_1, \widetilde{\mathbf{u}}_1)| \lesssim \frac{C_u \, \epsilon}{c(A, B) \sin(\phi_1 - \phi_2)}. \tag{6}$$

Although perturbation upper bound has been studied extensively, there are few results on lower bound especially in algebra literature. Next we establish the perturbation lower bound. That is, given any symmetric definite matrix pair $(A, B)$ with $\lambda_1 > \lambda_2$ and any sufficiently small constant $\epsilon > 0$, we can always find a matrix pair $(E, F)$ satisfying $\sqrt{\|E\|_2^2 + \|F\|_2^2} < \epsilon$ such that the distance between $\mathbf{u}_1$ and $\widetilde{\mathbf{u}}_1$ is lower bounded by a quantity in the same order of the bound in Lemma 1. The result is stated in the next lemma.

**Lemma 2** *Follow the notations in Lemma 1. For any small positive constant* $\epsilon$, *it holds that*

$$\sup_{(E, F) \in \mathcal{F}_\epsilon} |\sin \theta(\mathbf{u}_1, \widetilde{\mathbf{u}}_1)| \gtrsim \frac{C_l \, \epsilon}{c(A, B) \sin(\phi_1 - \phi_2)}, \tag{7}$$

*where* $C_l = \frac{c(A, B)}{\sqrt{2(\|A\|_2^2 + \|B\|_2^2) \kappa^{\frac{3}{2}}}}$ *with* $\kappa := \|B\|_2 \|B^{-1}\|_2$, $\mathcal{F}_\epsilon := \{(E, F) | \sqrt{\|E\|_2^2 + \|F\|_2^2} \leq \epsilon\}$.

Comparing (6) and (7) and noticing that $C_u$, $C_l$ are two constants, we can see that upper and lower bounds of approximation error only differ up to a multiplicative constant when condition number $\kappa$ is assumed to be bounded. Quantity $\epsilon/c(A, B)$ can be seen as the relative perturbation and $\frac{1}{\sin(\phi_1 - \phi_2)}$ is a monotonically decreasing function of the gap $\phi_1 - \phi_2$. The upper and lower bounds are both proportional to $\frac{\epsilon}{c(A, B) \sin(\phi_1 - \phi_2)}$. Thus, we declare that the perturbation bound for the leading eigenvector $\mathbf{u}_1$ is *rate-optimal*.

### 3.2 On the Sparse Setting

Now we are ready to present the upper and lower perturbation bounds under the sparse setting. Throughout the rest of this section, the following assumptions are assumed.

**A1** For any $\mathcal{K} \supset \text{supp}(\mathbf{u}_1)$ with $|\mathcal{K}| \leq s + k$, it holds

$$\frac{\sqrt{\|\widetilde{A}_{\mathcal{K}} - A_{\mathcal{K}}\|_2^2 + \|\widetilde{B}_{\mathcal{K}} - B_{\mathcal{K}}\|_2^2}}{c(A_{\mathcal{K}}, B_{\mathcal{K}})} < 1,$$

where $c(A_{\mathcal{K}}, B_{\mathcal{K}})$ is the Crawford number of $(A_{\mathcal{K}}, B_{\mathcal{K}})$ and $s = |\text{supp}(\mathbf{u}_1)|$.

**A2** For any $\mathcal{K} \supset \text{supp}(\mathbf{u}_1)$ with $|\mathcal{K}| \leq s + k$, $\widetilde{A}_{\mathcal{K}}$ and $\widetilde{B}_{\mathcal{K}}$ are positive definite.

Assumption **A1** requires that the perturbation within a small superset of $\mathbf{u}_1$ is tiny. It says that one can get a good approximation for $\mathbf{u}_1$ (according to Lemma 1 and 2) when a small superset of $\mathbf{u}_1$ is available. It is, in fact, a necessary condition for computing a good approximation for $\mathbf{u}_1$. Assumption **A2** is a technical requirement to ensure positive definiteness for submatrices of $\tilde{B}$.

We further adopt the following notations. We let $\mathcal{S} := \text{supp}(\mathbf{u}_1)$ and define the perturbation level

$$\epsilon := \max_{\mathcal{K}: \mathcal{K} \supset \mathcal{S}; |\mathcal{K}| \leq s+k} \sqrt{\|\widetilde{A}_{\mathcal{K}} - A_{\mathcal{K}}\|_2^2 + \|\widetilde{B}_{\mathcal{K}} - B_{\mathcal{K}}\|_2^2},$$

the perturbation set $\mathcal{F}_{\epsilon,l} := \{(E, F) | \sqrt{\|E_{\mathcal{K}}\|_2^2 + \|F_{\mathcal{K}}\|_2^2} \leq \epsilon$ for any subset $\mathcal{K}$ with size less than $l\}$.

**Upper Bound of** $|\sin\theta(\mathbf{u}_1, \mathbf{x}_*)|$

**Theorem 1** *Let* $\mathcal{J} = \text{supp}(\mathbf{x}_*)$, $\mathcal{K} = \mathcal{S} \cup \mathcal{J}$ *and* $= \mathcal{K} \setminus \mathcal{J}$. *Denote* $\rho_* = \frac{\mathbf{x}_*^{\mathsf{T}} \widetilde{A} \mathbf{x}_*}{\mathbf{x}_*^{\mathsf{T}} \widetilde{B} \mathbf{x}_*}$, $c_{\mathcal{K}} = c(A_{\mathcal{K}}, B_{\mathcal{K}})$ *and* $\widetilde{c}_{\mathcal{K}} = c(\widetilde{A}_{\mathcal{K}}, \widetilde{B}_{\mathcal{K}})$. *Let* $\delta = \frac{\|\widetilde{A}_{(,\mathcal{J})}[\mathbf{x}_*]_{\mathcal{J}} - \rho_* \widetilde{B}_{(,\mathcal{J})}[\mathbf{x}_*]_{\mathcal{J}}\|_2}{\widetilde{c}_{\mathcal{K}}}$. *If* $\delta < \sqrt{1 + \rho_*^2}$ *and*

$$\arctan \rho_* > \arctan \mu_2 + \arctan \epsilon + \arctan \frac{\delta}{\sqrt{1 + \rho_*^2}}, \tag{8}$$

*then*

$$|\sin\theta(\mathbf{u}_1, \mathbf{x}_*)| \leq \frac{1}{\widetilde{c}_{\mathcal{K}}} \left( \frac{\epsilon\, C_\epsilon}{c(A_{\mathcal{K}}, B_{\mathcal{K}}) \sin(\phi_1 - \widetilde{\phi}_2)} + \frac{\delta\, C_\delta}{\sin(\phi_* - \widetilde{\phi}_2)} \right),$$

*where* $\phi_* = \arctan \rho_*$, $\mu_2$ *is the second largest eigenvalue for* $(A_{\mathcal{K}}, B_{\mathcal{K}})$, $C_\epsilon$ *and* $C_\delta$ *are some constants.*

**Remark**: The upper bound has two terms. The first term is due to the perturbation, which is approximately proportional to the perturbation level $\epsilon$. The second term is due to the failure in finding the true support set of $\mathbf{u}_1$. If $\text{supp}(\mathbf{u}_1) \subset \text{supp}(\mathbf{x}_*)$, then $\delta = 0$, consequently, the second term vanishes. Therefore, it holds asymptotically that

$$|\sin\theta(\mathbf{u}_1, \mathbf{x}_*)| \leq \frac{C_{u,\mathcal{K}}\, \epsilon}{c(A_{\mathcal{K}}, B_{\mathcal{K}}) \sin(\phi_1 - \phi_2)} \tag{9}$$

with $C_{u,\mathcal{K}} = \frac{\sqrt{2(\|A_{\mathcal{K}}\|_2^2 + \|B_{\mathcal{K}}\|_2^2)}}{c(A_{\mathcal{K}}, B_{\mathcal{K}})}$, when $\text{supp}(\mathbf{x}_*) \supset \text{supp}(\mathbf{u}_1)$ and $\epsilon$ is sufficiently small.

**Lower Bound of** $|\sin\theta(\mathbf{u}_1, \mathbf{x}_*)|$   Next we present the lower bound for $|\sin\theta(\mathbf{u}_1, \mathbf{x}_*)|$. It follows immediately from the proofs of Theorem 1 and Lemma 2.

**Theorem 2** *Follow the notations and assumptions in Theorem 1. Then the following result holds.*

**Case (a)**  *If* $\delta = 0$*, then*

$$\max_{(E,F)\in\mathcal{F}_{\epsilon,s+k}} |\sin\theta(\mathbf{u}_1, \mathbf{x}_*)| \gtrsim \frac{C_{l,\mathcal{K}}\, \epsilon}{c(A_{\mathcal{K}}, B_{\mathcal{K}}) \sin(\phi_1 - \phi_2)},$$

*where* $C_{l,\mathcal{K}} = \frac{c(A_{\mathcal{K}}, B_{\mathcal{K}})}{\sqrt{2(\|A_{\mathcal{K}}\|_2^2 + \|B_{\mathcal{K}}\|_2^2)} \kappa_2^{\frac{3}{2}}(B_{\mathcal{K}})}$.

**Case (b)**  *If* $\epsilon = 0$*, then*

$$\max_{(E,F)\in\mathcal{F}_{\epsilon,s+k}} |\sin\theta(\mathbf{u}_1, \mathbf{x}_*)| \gtrsim \frac{1}{\sqrt{2}\kappa_2^{\frac{3}{2}}(B_{\mathcal{K}})} \frac{\widehat{\xi}_{\mathcal{K}}}{\sin(\phi_* - \phi_2)},$$

*where* $\widehat{\xi}_{\mathcal{K}} = \sqrt{\frac{\|\widehat{E}[u_1]_{\mathcal{K}}\|_2^2 + \|\widehat{F}[u_1]_{\mathcal{K}}\|_2^2}{\|A\|_2^2 + \|B\|_2^2}}$, *and* $\widehat{E}, \widehat{F} \in \mathbb{R}^{|\mathcal{K}| \times |\mathcal{K}|}$ *are some matrices controlled by* $\delta$.

Note that Crawford and condition number is well bounded on submatrix pair $(\mathcal{A}_\mathcal{K}, \mathcal{B}_\mathcal{K})$. Therefore, in **Case (a)**, we can see that the lower bound is at the same order of the first term of the upper bound in Theorem 1; In **Case (b)**, $\widehat{\xi}_\mathcal{K} = \mathcal{O}(\delta)$, then the lower bound is at the same order of the second term of the upper bound in Theorem 1. We may declare that our perturbation bounds are *rate-optimal*.

**Connection to the literature**   Although we do not make any statistical assumptions on $E$ and $F$, we still make connections to the statistical literature here. First, it can be computed that

$$\sin(\phi_1 - \phi_2) = \frac{\lambda_1 - \lambda_2}{\sqrt{\lambda_1^2 + 1}\sqrt{\lambda_2^2 + 1}}.$$

When $B = I$ is an identity matrix, $C_{u,\mathcal{K}} = O(\sqrt{\lambda_1^2 + 1})$ and $C_{l,\mathcal{K}} = \Omega(\frac{1}{\sqrt{\lambda_1^2+1}})$. Thus our lower bound can be simplified to $\frac{\sqrt{\lambda_2^2+1}}{\lambda_1 - \lambda_2}\epsilon$. In sparse PCA [36], it has been established that the minimax lower bound $\inf_{\tilde{\mathbf{u}}_1} \sup_{\mathcal{M}}(\mathbb{E}\|\mathbf{u}_1 - \tilde{\mathbf{u}}_1\|_2^2)^{1/2}$ is $\Theta(\frac{\sqrt{\lambda_1 \lambda_2}}{\lambda_1 - \lambda_2}\epsilon)$ with $\epsilon = \sqrt{\frac{s \log p}{n}}$ and $\mathcal{M}$ is the model space where covariance matrix $A$ satisfies that its principle singular vector is $s$-sparse and $\lambda_1$ is larger than $\lambda_2$ by certain margin. When $\lambda_1, \lambda_2$ and $\lambda_1/\lambda_2$ are bounded constants, this minimax rate matches ours. In [2], they provide the minimax perturbation bound for singular subspace of non-symmetric matrices. The minimax rate of $\|\tilde{\mathbf{u}}_1 - \mathbf{u}_1\|_2$ is

$$\Theta(\frac{\alpha z_{21} + \beta z_{12}}{\alpha^2 + \beta^2 - \min\{z_{12}^2, z_{21}^2\}}),$$

where the detailed definitions of $\alpha$, $\beta$, $z_{12}$, $z_{21}$ can be found accordingly in [2]. Especially, for symmetric $A$ and $\tilde{A}$, the above bound can be simplified to $\frac{\epsilon}{\lambda_1 - \lambda_2}$ when $\epsilon \ll \lambda_1 - \lambda_2$. Again, when $\lambda_1$ and $\lambda_2$ are bounded constants, the rate is consistent with ours.

## 4   Understanding of Estimation Quality

In order to estimate the sparse leading eigenvector, it is straightforward to solve (1). It can be seen that the sparsity of solution to (1) depends on the choice of $k$, which may not be easily tuned. It is also computationally expensive to find the optimal solution to (1). Alternatively, such problem can be solved via regularization method. For example, [23] uses a smooth surrogate function to replace $\|\cdot\|_0$ penalty for solving SGEP.

A good estimator should have the following properties, 1. sparsity, 2. nearly unbiasedness, and 3. stability. In high dimensional statistical problem, $\tilde{B}$ is always singular when the sample size is smaller than the number of features. In order to obtain a good approximation of $\mathbf{u}_1$, we consider the following restricted problem

$$(P1') \quad \min_{\mathbf{x} \in \mathbb{R}^p, \|\mathbf{x}\|_0 \le s_n} -\mathbf{x}^\mathrm{T}\widetilde{A}\mathbf{x} + p_\lambda(\mathbf{x}) \qquad s.t. \ \mathbf{x}^\mathrm{T}\widetilde{B}\mathbf{x} \le 1, \tag{10}$$

where $p_\lambda(\mathbf{x}) := \sum_{j=1}^p p_\lambda(\mathbf{x}[j])$ and $p_\lambda(x)$ is some univariate non-convex function, and $s_n$ is the restricted dimension. In practice, $s_n$ need not be a small number and it can grow with sample size $n$. The restriction $\|\mathbf{x}\|_0 \le s_n$ is imposed for enforcing the solution to be nearly low-dimensional. Along with $p_\lambda$, the estimator can recover the sparsity structure. This reformulation can be viewed as the counterpart of two stage methods [30, 9], where they need to find a good approximation of $\mathbf{u}_1$ in the first stage.

### 4.1   Penalization Function

For the choice of $p_\lambda$, we consider a family of special non-convex penalties:

$$\mathcal{P}_\lambda = \{p_\lambda(x) : p_\lambda(x) \text{ satisfies (a1) - (a3)}\}, \tag{11}$$

where (a1) Function $p_\lambda(x)$ is an even function, i.e., $p_\lambda(x) = p_\lambda(-x)$; (a2) The derivative of $p_\lambda(x)$, $p_\lambda'(x)$, exists in $(0, \infty)$; $\lim_{x\downarrow 0} p_\lambda'(x) = \lambda$ and $p_\lambda'(x) \equiv 0$ if $x \ge \gamma\lambda$ for some constant $\gamma$; (a3) On $(0, \infty)$, $p_\lambda'(x)$ is monotone decreasing and Lipschitz continuous, i.e., there exists a constant $\kappa$ such that $0 \le \frac{p_\lambda'(x_1) - p_\lambda'(x_2)}{x_2 - x_1} \le \kappa$ for any $0 < x_1 < x_2$.

Here (a1) requires $p_\lambda(x)$ to be symmetric; (a2) specifies the local property of derivatives of $p_\lambda(x)$ around 0 and assumes the flatness of $p_\lambda(x)$ for larger $x$; (a3) puts continuity constraints on $p'_\lambda(x)$. Many popular penalty functions are included in $\mathcal{P}_\lambda$, for example, smoothly clipped absolute deviation (**SCAD**, [7]), minimax concave penalty (**MCP**, [43]) , etc. Based on the definitions, we can see that any $p_\lambda(x) \in \mathcal{P}_\lambda$ is similar to $\ell_1$-norm locally around the origin. On the other hand, $p_\lambda(x)$ puts a smaller penalization on the signal compared with $\ell_1$ penalty. They do not give the penalization to those large values. Intuitively speaking, the regularized estimator with non-convex penalty may outperform the $\ell_1$-norm-based estimator since it is nearly unbiased.

## 4.2 Support Recovery

In this section, we provide the error bound and oracle properties of the non-convex estimator under suitable conditions. Some additional notations are introduced as follows. We define $\hat{\mathbf{u}}_1$ as $\arg\max_{\mathbf{x}:\|\mathbf{x}\|_0 \le s_n, \mathbf{x}^{\mathrm{T}}\widetilde{B}\mathbf{x} \le 1} \mathbf{x}^{\mathrm{T}}\tilde{A}\mathbf{x}$ and denote $\tilde{\rho} = \hat{\mathbf{u}}_1^{\mathrm{T}}\tilde{A}\hat{\mathbf{u}}_1$. Vector $\hat{\mathbf{u}}_1$ can be viewed as the best approximation of the leading eigenvector in the restricted space. We define $c(A, B, s_n) := \max_{|\mathcal{K}| \le s_n} c(A_\mathcal{K}, B_\mathcal{K})$ and $c(\tilde{A}, \widetilde{B}, s_n) := \max_{|\mathcal{K}| \le s_n} c(\widetilde{A}_\mathcal{K}, \widetilde{B}_\mathcal{K})$. We also define $N(A, B, s_n) := \max_{|\mathcal{K}| \le s_n} \sqrt{\|A_\mathcal{K}^2 + B_\mathcal{K}^2\|_2}$. We define $\epsilon_s := \max_{\mathcal{K}:\|\mathcal{K}\|_0 \le s_n} \{\|E_\mathcal{K}\|_2, \|F_\mathcal{K}\|_2\}$. A different set of assumptions is stated as follows.

- **B0** (*Regularity*) $\lambda_1 - \lambda_2$ is positive; $\|B\|, \|B^{-1}\|$ are bounded by some constant.
- **B1** (*Signal*) $\min\{|\mathbf{u}_1[j]| : j \in \text{supp}(\mathbf{u}_1)\} \gg \sqrt{S}\lambda$.
- **B2** (*Penalization*) $\lambda \gg \epsilon_s \cdot C_F$, where $C_F$ is a constant depending on $c(A, B, s_n), c(\tilde{A}, \tilde{B}, s_n)$ and $N(A, B, s_n)$.
- **B3** (*Support size*) $|\mathcal{S}|\lambda^2 \ll 1$.
- **B4** (*Identifiability*) $\max_{\mathcal{K}:\|\mathcal{K}\|_0 \le s_n, \mathcal{S} \not\subset \mathcal{K}} \lambda_1(A_\mathcal{K}, B_\mathcal{K}) < \lambda_1 - 2\epsilon$.

Condition **B0** ensures that the leading eigenvector is unique and underlying matrix $B$ is well-behaved. Condition **B1** requires that the absolute values of entries in the true support is not too small so that $\mathcal{S}$ can be identified. Condition **B2** specifies the relationship between penalty level $\lambda$ and noise level $\epsilon$. That is, penalty level $\lambda$ should be at least larger than the noise level up to a multiplicative constant. Condition **B3** makes sure that the support size is not too large, i.e., $\mathbf{u}_1$ should be sparse. The following theorem guarantees that the estimator is not far away from the re-scaled leading eigenvector. Condition **B4** is for the identifiability of $|\mathcal{S}|$ to ensure that the support of $\mathbf{u}_1$ could be still identified after perturbation. Without loss of generality, we always assume $s_n \ge |\mathcal{S}|$. Then we have the following results.

**Theorem 3** *Let $\hat{\mathbf{x}}$ be the optimizer of Problem (P1'). Under Conditions **B0** - **B4**, it holds that*
$$\min_{\text{sgn}\in\{-1,1\}} \|\text{sgn} \cdot \hat{\mathbf{x}} - \mathbf{u}_{1s}\|_2 \le C_1 \sin(\theta(\hat{\mathbf{u}}_1, \mathbf{u}_1)) + C_2 \epsilon_s,$$
*for some constants $C_1$ and $C_2$.*

Here $\mathbf{u}_{1s} = \mathbf{u}_1(\mathbf{u}_1^{\mathrm{T}}B\mathbf{u}_1)^{-1/2}$ is the re-scaled leading eigenvector. Furthermore, we can recover the support of $\mathbf{u}_1$ under this restricted problem.

**Theorem 4** *Under the same conditions in Theorem 3, it holds that*
$$\text{supp}(\hat{\mathbf{x}}) = \text{supp}(\mathbf{u}_1). \tag{12}$$

Based on Theorem 4, we actually have even stronger results. When all required conditions are met, the proposed estimator is equal to the oracle estimator which is the one estimated when the true support $\mathcal{S}$ is known to us. Therefore, the estimator achieves the optimal error bound which matches the one obtained in Theorem 1. Specifically, the proposed estimator achieves the optimal error rate $(\frac{|\mathcal{S}|\log p}{n})^{1/2}$ in sparse PCA and sparse CCA problems.

**Theorem 5** *Under the same set of conditions in Theorem 4, we have that*
$$|\sin\theta(\hat{\mathbf{x}}, \mathbf{u}_1)| \le C_{u,\mathcal{S}} \frac{\epsilon}{c(A_\mathcal{S}, B_\mathcal{S})\sin(\phi_1 - \phi_2)}, \tag{13}$$
*with $C_{u,\mathcal{S}} = C\frac{\sqrt{2(\|A_\mathcal{S}\|_2^2 + \|B_\mathcal{S}\|_2^2)}}{c(A_\mathcal{S}, B_\mathcal{S})}$ (C is a universal constant).*

**Remark 1** *To achieve the optimal statistical error rate $(\frac{|\mathcal{S}|\log p}{n})^{1/2}$, most existing methods are two-stage based. For example, [38] studies sparse PCA problem and propose a "sparse orthogonal iteration pursuit" (SOAP) algorithm which involves "relax" and "tighten" stages; [9] considers sparse CCA problem and adopt the two-stage regularization approach with L1 penalty for the first stage and group LASSO penalty for the second stage. By comparison, our current estimator does not require convex relaxation in the first stage.*

## 4.3 Computation

For computational purpose, we consider to solve (P1') by using the alternating direction method of multipliers (ADMM, [37, 21]). Specifically, we relax the problem (P1') by reformulating it to

$$\min_{\mathbf{x},\mathbf{y},\mathbf{z}} \mathcal{L}(\mathbf{x},\mathbf{z},\mathbf{y}), \qquad s.t. \ \mathbf{x}^{\mathrm{T}}\widetilde{B}\mathbf{x} = 1, \ \|\mathbf{x}\|_0 \le s_n, \tag{14}$$

where

$$\mathcal{L}(\mathbf{x},\mathbf{z},\mathbf{y}) = -\mathbf{x}^{\mathrm{T}}\tilde{A}\mathbf{x} + p_\lambda(\mathbf{z}) + \mathbf{y}^{\mathrm{T}}(\mathbf{x}-\mathbf{z}) + \frac{\eta}{2}\|\mathbf{x}-\mathbf{z}\|_2^2.$$

In (14), we introduce several auxiliary variables for the following reasons. We construct a new vector $\mathbf{z}$ which is a copy of $\mathbf{x}$. It can help us to split the original problem to two simple separate sub-problems. $\mathbf{y}$ is the dual variable for the constraint $\mathbf{x} = \mathbf{z}$.

By formulation (14), we can optimize the objective function with respect to each variable iteratively. The steps for updating $\mathbf{x}, \mathbf{y}, \mathbf{z}$ are described as follows.

    i *Update* $\mathbf{x}$: At $(t+1)$-th iteration, we aim to find $\mathbf{x}^{(t+1)}$ which is

$$\arg\min_{\mathbf{x}\in\mathcal{D}} \quad \frac{\eta}{2}\|\mathbf{x}-\mathbf{z}^{(t)}\|_2^2 + (\mathbf{y}^{(t)})^{\mathrm{T}}(\mathbf{x}-\mathbf{z}^{(t)}) - \mathbf{x}^{\mathrm{T}}\tilde{A}\mathbf{x}, \tag{15}$$

    where $\mathcal{D} = \{\mathbf{x} : \mathbf{x}^{\mathrm{T}}\widetilde{B}\mathbf{x} = 1, \|\mathbf{x}\| \le s_n\}$. Solve sub-problem (15) to get $\mathbf{x}^{(t+1)}$.

    ii *Update* $\mathbf{z}$: We know that $\mathbf{z}^{(t+1)} = \arg\min_{\mathbf{z}} \frac{\eta}{2}\|\mathbf{z}-\mathbf{x}^{(t)}\|_2^2 + (\mathbf{y}^{(t)})^{\mathrm{T}}(\mathbf{x}^{(t+1)}-\mathbf{z}) + p_\lambda(\mathbf{z})$. Each entry of $\mathbf{z}$ can be optimized separately. Specifically, if we take $p_\lambda$ as the MCP penalty, then $\mathbf{z}^{(t+1)}$ has the following analytical form,

$$\mathbf{z}^{(t+1)}[j] = \begin{cases} \check{\mathbf{z}}^{(t)} & \text{if } |\check{\mathbf{z}}^{(t)}| > \gamma\lambda, \\ \frac{\text{sgn}(\check{\mathbf{z}}^{(t)}[j])(|\check{\mathbf{z}}^{(t)}[j]|-\frac{\lambda}{\eta})_+}{1-\frac{1}{\eta\gamma}} & \text{if } |\check{\mathbf{z}}^{(t)}| \le \gamma\lambda, \end{cases}$$

    where $\check{\mathbf{z}}^{(t)} := \mathbf{x}^{(t+1)} + \frac{\mathbf{y}^{(t)}}{\eta}$.

    iii *Update* $\mathbf{y}$: By dual variable update in [21], we have $\mathbf{y}^{(t+1)} = \mathbf{y}^{(t)} + \eta(\mathbf{x}^{(t+1)} - \mathbf{z}^{(t+1)})$.

We call the above procedure as non convex-SGEP (NC-SGEP) algorithm. Such proposed algorithm works on matrix-vector product and eigen-decomposition for submatrices of $\tilde{B}$, which is computationally efficient. Compared with other truncation methods, we do not need to make efforts to choose best $s_n$ due to the existence of non-convex regularization term. Thanks to the penalization term, our method can give a sparse estimator with a very wide range to choose the restricted dimension $s_n$.

Here we propose a *projection*-based method for solving sub-problem (15). At $(t+1)$-th iteration, we aim to find $\mathbf{x}^{(t+1)}$ which is the minimizer of (15). We construct the active set $\mathcal{S}_t$ and consider the following recursive formula,

$$\begin{aligned} \mathbf{b}_m^{(t+1)} &= \mathbf{z}^{(t)} - \frac{\mathbf{y}^{(t)} - \tilde{A}\mathbf{x}_{m-1}^{(t+1)}}{\eta}, \\ \mathbf{x}_m^{(t+1)}[\mathcal{S}_t] &= (\beta_m^{(t+1)}\widetilde{B}_{\mathcal{S}_t} + I)^{-1}\mathbf{b}_m^{(t+1)}[\mathcal{S}_t], \end{aligned} \tag{16}$$

where $\mathbf{x}_0^{(t+1)} = \mathbf{x}^{(t)}$ and index $m \in \{1,2,\ldots\}$. Scalar $\beta_m^{(t+1)}$ satisfies $1 = \sum_j \frac{d_j(\widetilde{\mathbf{b}}_m^{(t+1)}[j])^2}{(\beta_m^{(t+1)}d_j+1)^2}$ with $\widetilde{\mathbf{b}}_m^{(t+1)} = U^{\mathrm{T}}\mathbf{b}_m^{(t+1)}[\mathcal{S}_t]$; $\widetilde{B}_{\mathcal{S}_t} = UDU^{\mathrm{T}}$ and $D = \text{diag}(d_1,\ldots,d_{s_n})$. Such recursive formula is valid due to the following two observations (Propositions 1 - 2).

**Proposition 1** *Let $\check{\mathbf{y}}$ be the projection of $\mathbf{y}$ on to the ellipsoid $\{\mathbf{x} \mid \mathbf{x}^{\mathrm{T}} B \mathbf{x} = 1\}$. Then $\check{\mathbf{y}}$ has the following form*

$$\check{\mathbf{y}} = (\beta B + I)^{-1} \mathbf{y},$$

*where $\beta$ is a scalar which is the solution to the equation $1 = \sum_j \frac{d_j (\tilde{\mathbf{y}}[j])^2}{(\beta d_j + 1)^2}$, where $\tilde{\mathbf{y}} = U^{\mathrm{T}} \mathbf{y}$, $B = U D U^{\mathrm{T}}$ and $D = \mathrm{diag}(d_1, \ldots, d_p)$.*

**Proposition 2** *The limiting point returned by* (16) *is the stationary point of* (15).

Proposition 1 gives the explicit formula to project an arbitrary vector $\mathbf{y}$ to the convex body $\{\mathbf{x} | \mathbf{x}^{\mathrm{T}} B \mathbf{x} = 1\}$. By contrast, it may lead to worse performance, if we just do the naive rescaling method, i.e., $\check{\mathbf{y}} = \mathbf{y}(\mathbf{y}^{\mathrm{T}} B \mathbf{y})^{-1/2}$. Since (15) is highly non-convex, Proposition 2 only guarantees a way to find a stationary solution but not necessarily a optimal solution.

In this paper, we consider the following two possible constructions of active set $\mathcal{S}_t$.

- (C1) $\mathcal{S}_t$ is the set of indices corresponding to first $s_n$ largest absolute values of entries in $\mathbf{b}_1^{(t+1)}$ (see (16)).
- (C2) $\mathcal{S}_t$ is the set of indices corresponding to first $s_n$ largest absolute values of entries in $\mathbf{x}^{(t)}$.

The following theorem gives the local convergence of NC-SGEP algorithm under additional assumptions. In general, ADMM-based method is extremely hard to analyze especially in highly non-convex problem. It remains an open question whether the global convergence result could be established.

**Theorem 6** *For the active set construction (C2), we take a large $\eta$ value, set the initial support of $\mathbf{x}^{(0)}$ includes $\mathrm{supp}(\mathbf{u}_1)$ and let $\mathbf{z}^{(0)} = \mathbf{x}^{(0)}, \mathbf{y}^{(0)} = \mathbf{0}$. Then it holds*

$$T(\epsilon) \leq \frac{C(\mathcal{L}(\mathbf{x}^{(0)}, \mathbf{z}^{(0)}, \mathbf{y}^{(0)}) - \bar{f})}{\epsilon},$$

*where $T(\epsilon) = \min\{t : \|\mathbf{x}^{(t)} - \mathbf{z}^{(t)}\| \leq \epsilon\}$ and $\bar{f} = \min_{\mathbf{x}} \mathcal{L}(\mathbf{x}, \mathbf{x}, \mathbf{y})$ for some constant $C$ which may depend on $\eta$ and $\max_{\mathcal{S}:|S| \leq s_n} \|(B_{\mathcal{S}})^{-1/2} A_{\mathcal{S}} (B_{\mathcal{S}})^{-1/2}\|_2$. Moreover, $\mathrm{supp}(\mathbf{x}^{(T(\epsilon))}) = \mathrm{supp}(\mathbf{u}_1)$ if $\epsilon = o(\epsilon_s)$.*

For construction method (C1), the same local convergence result also holds with an extra assumption that $\tilde{B}$ is diagonal.

### 4.4 Remarks

In practice, we find that construction (C1) has higher probabilities to find the global optimum compared with construction (C2). Especially in the application of sparse principle component analysis, construction (C1) is very efficient to recover the sparsity structure of leading component. Theorem 6 gives a local convergence result, which requires the initial value $\mathbf{x}^{(0)}$ contains the support of true leading eigenvector $\mathbf{u}_1$. Such a good initial candidate of leading eigenvector could be obtained via using semidefinite programming (SDP)-based methods [35, 36, 9]. In other words, our proposed method can be easily merged to a two-stage-type method. The proposed algorithm only requires $O(s_n p + s_n^3)$ operations per iteration, while SDP-based methods have $O(p^3)$ computational complexity. More discussions about SDP-based methods can be found in the supplemental material.

## 5 Numerical Experiments

### 5.1 Validation of Perturbation Bounds

We conduct perturbation analyses of proposed estimator under different settings. The matrix pair is set as $A = 4\mathbf{u}_1 \mathbf{u}_1^{\mathrm{T}} + I - P_{\mathbf{u}_1}$, $B = I$. We sample $n$ data which follows $N(0, A)$ and sample another $n$ data which follows $N(0, B)$. The $\tilde{A}$ and $\tilde{B}$ are constructed based on the sample covariance correspondingly. The leading eigenvector $\mathbf{u}_1$ has unit norm and has non-zero entries in first $|\mathcal{S}|$ positions. We fixed dimension $p \equiv 100$ and let number of samples ($n$) and support size ($|\mathcal{S}|$) vary. Each setting is repeated for 100 times with fixed choice of $\lambda = 0.3, \eta = 1, s_n = 25$. The mean and

Table 1: Estimation error $\|\hat{\mathbf{x}} - \mathbf{u}_1\|_2$ under different perturbation and sparsity level. "Oracle" / "Est": with / without knowing support $\mathcal{S}$. "Sd" is the standard deviation of "Est".

| | $n$ | 100 | 200 | 400 | 800 | 1600 | 3200 | 6400 |
|---|---|---|---|---|---|---|---|---|
| $|\mathcal{S}|$ | $\sqrt{\frac{\log p}{n}}$ | 0.230 | 0.162 | 0.115 | 0.081 | 0.057 | 0.041 | 0.029 |
| 2 | Oracle | 0.056 | 0.036 | 0.029 | 0.019 | 0.013 | 0.010 | 0.007 |
| | Est | 0.140 | 0.047 | 0.043 | 0.024 | 0.019 | 0.012 | 0.008 |
| | Sd | (0.190) | (0.034) | (0.030) | (0.022) | (0.013) | (0.009) | (0.006) |
| 4 | Oracle | 0.107 | 0.073 | 0.057 | 0.037 | 0.028 | 0.019 | 0.014 |
| | Est | 0.225 | 0.131 | 0.077 | 0.043 | 0.032 | 0.022 | 0.017 |
| | Sd | (0.244) | (0.061) | (0.036) | (0.028) | (0.017) | (0.011) | (0.009) |
| 8 | Oracle | 0.173 | 0.118 | 0.087 | 0.062 | 0.044 | 0.030 | 0.022 |
| | Est | 0.277 | 0.191 | 0.110 | 0.082 | 0.058 | 0.043 | 0.028 |
| | Sd | (0.279) | (0.156) | (0.066) | (0.039) | (0.031) | (0.022) | (0.012) |

standard deviation of $\|\hat{\mathbf{x}} - \mathbf{u}_1\|_2$ are reported. We also compute the oracle estimator which is the best $\hat{\mathbf{x}}$ when true support $\mathcal{S}$ is known. From Table 1, when noise level $\sqrt{\log p/n}$ is small, we can see that the estimation error is quite close to the optimal (oracle) error. In addition, we can see that estimation error is proportional to $\sqrt{|\mathcal{S}|}$ and $\sqrt{\log p/n}$. This indicates that our method can achieve the theoretical optimal error rate, i.e., $O(\sqrt{\frac{|\mathcal{S}|\log p}{n}})$.

## 5.2 Validation of Sparsity Recovery

The underlying matrix pair is set as $A = 3\mathbf{u}_1\mathbf{u}_1^{\mathrm{T}} + I - P_{\mathbf{u}_1}$, $B = I$. Thus, we naturally take $\widetilde{B} = B = I$ and $\widetilde{A}$ as the sample covariance of data. We let dimension $p$ grow from 16 to 256, fix the sample size $n = 100$ and set $\lambda = 0.5$, $\eta = 1$ and $s_n = 50$. We compare the proposed method with the semidefinite programming method [34, 18, 40, SDP] with $\ell_1$ (SDP_L1) and MCP (SDP_MCP) penalty. The estimation error and percentage of support recovery are reported in Table 2. We can see that the proposed method can have a slightly better performance. This is because that NC-SGEP optimizes objective within space $\mathbb{R}^p$ unlike those SDP methods work on space $\mathbb{R}^{p \times p}$. In addition, the proposed method can recover the sparsity structure pretty well, while SDP methods can never recover the true support set.

Table 2: Estimation accuracy for sparse canonical correlation analysis.

| | $p$ | 16 | 32 | 64 | 128 | 256 |
|---|---|---|---|---|---|---|
| | **NC-SGEP** | 0.074 (0.056) | 0.076 (0.068) | 0.072 (0.059) | 0.081(0.054) | 0.084 (0.054) |
| $\|\hat{\mathbf{x}}\hat{\mathbf{x}}^{\mathrm{T}} - \mathbf{u}_1\mathbf{u}_1^{\mathrm{T}}\|_\infty$ | SDP_L1 | 0.071 (0.046) | 0.080 (0.041) | 0.095 (0.049) | 0.094 (0.039) | 0.100 (0.041) |
| | SDP_MCP | 0.070 (0.045) | 0.079 (0.040) | 0.093 (0.048) | 0.093 (0.038) | 0.096 (0.039) |
| | NC-SGEP | 93 % | 82 % | 68 % | 59 % | 49 % |
| Recovery of $|\mathcal{S}|$ | SDP_L1 | - | - | - | - | - |
| | SDP_MCP | - | - | - | - | - |

## 6 Conclusion

In this paper, we establish the upper and lower bounds for perturbation analysis of sparse generalized eigenvalue problem. We also consider a new statistical estimation method. The proposed method gives a sparse, nearly unbiased and stable solution to SGEP. We show that the proposed estimator can achieve the optimal estimation error rate. We further present a non-convex SGEP (NC-SGEP) algorithm to solve a non-convex regularization problem with guarantee of local convergence. Multiple numerical results validate our theories and also show the superior performance of the proposed method. In the future work, on the theoretical side, we may focus on extending the current results to the problem of finding multiple leading sparse eigenvectors and establishing the corresponding new lower bound theory. On the computational side, even though computing leading vectors one by one seems straightforward, it is not an easy problem since it requires the orthogonalization procedure. The computational complexity may require a special care.

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
