# A Note on Sparse Generalized Eigenvalue Problem

**Yunfeng Cai, Guanhua Fang, Ping Li**
Cognitive Computing Lab
Baidu Research
No. 10 Xibeiwang East Road, Beijing 100193, China
10900 NE 8th St. Bellevue, Washington 98004, USA
{yunfengcai, guanhuafang, liping11}@baidu.com

**Outline of Supplementary Material.** In Section 7, we provide a connection between the proposed method and semidefinite programming-based methods. In Section 8, additional simulation results are provided. All technical proofs are presented in Section 9 - Section 14. Finally, the computational issues of semidefinite programming are discussed in Section 15.

## 7 Connection to Semidefinite Programming

For completeness, in this section, we provide discussions on semidefinite programming (SDP) method [34, 18, 40], which is usually well suited to solve regularization problems. It may relax the non-convex problem to a convex problem and can result in faster convergence. There exist a few literature on solving the eigenvector estimation problems via SDP. For example, [35, 36] study properties of SDP estimators in the SPCA setting. [9] proposes an efficient SDP method for estimating sparse canonical coefficients.

For SGEP, instead of directly penalizing the eigenvector, we can also recast the original problem into a semidefinite programming problem. We write $X = \mathbf{u}_1 \mathbf{u}_1^\mathrm{T}$. Then $\mathbf{u}_1^\mathrm{T} A \mathbf{u}_1$ is equal to $\mathrm{tr}(AX)$. For estimation, we consider the following SDP problem,

$$(P2) \quad \min_{X \in \mathbb{R}^{p \times p}} \quad -\mathrm{tr}(AX) + p_\lambda(X),$$
$$s.t. \quad \|B^{1/2} X B^{1/2}\|_2 \leq 1, \ \|B^{1/2} X B^{1/2}\|_* \leq r. \tag{17}$$

Here $p_\lambda(X) = \sum_{i,j} p_\lambda(X[i,j])$ and $r$ is taken to be 1. In the literature, the convex constraint $\mathcal{C}_{fan} := \{X \mid \|B^{1/2} X B^{1/2}\|_2 \leq 1, \|B^{1/2} X B^{1/2}\|_* \leq r\}$ is also known as a Fantope [8, 35]. Under this relaxed problem, we also provide the corresponding estimation error bound.

**Theorem 7** *Suppose $A$ and $B$ satisfy $A = B U_1 \Lambda_1 U_1^\mathrm{T} B + A_E$. Let $\widehat{X}$ be the optimizer of (P2). Then, as long as $\lambda \geq \sqrt{|\mathcal{S}|} \lambda_1 \|B^{-1/2}\| \|B^{1/2}\| (\|B\|_\infty + \|\widetilde{B}\|_\infty) \|\widetilde{B} - B\|_\infty + \|\widetilde{A} - A\|_\infty + \|A_E\|_\infty + e_{approx}$, then we have the following result.*

$$\|\widehat{X} - \mathbf{u}_1 \mathbf{u}_1^\mathrm{T}\|_F \leq C(B) \frac{|\mathcal{S}|\lambda}{\lambda_r}, \tag{18}$$

*where $U_1$ is the matrix of top $r$ eigenvector for $(A, B)$ and satisfies $U_1^\mathrm{T} B U_1 = I$; $\lambda_r$ is the $r$-th largest singular value; $C(B)$ is some constant depending on $B$ and $e_{approx}^2 := \max_{X \in \mathcal{C}_{fan}} \mathrm{tr}(\tilde{A}X) - \max_{X \in \mathcal{C}_{fan}, \|X\|_0 \leq |\mathcal{S}|^2} \mathrm{tr}(\tilde{A}X)$.*

In particular, if $A$ and $B$ are matrices with bounded entries and $\widetilde{A}$ and $\widetilde{B}$ are the corresponding sample version, then we know $\|\widetilde{A} - A\|_\infty \leq \sqrt{\frac{\log p}{n}}$ and $\|\widetilde{B} - B\|_\infty \leq \sqrt{\frac{\log p}{n}}$ hold with high probability. Term $e_{approx}$ is the price we need to pay since $p_\lambda$ puts the light penalization compared to usual $\ell_1$ norm especially when $\tilde{B}$ is singular. When $A$ admits a low rank structure and $(\tilde{A}, \tilde{B})$ admits a nearly sparse leading vector, we have the following corollary.

**Corollary 1** *Suppose we have two bounded matrices $A$ and $B$ satisfy $A = BU_1\Lambda_1 U_1^T B$. Let $\widetilde{A}$, $\widetilde{B}$ be the empirical estimates of $A$ and $B$ and $e_{approx} \leq C\|\tilde{A} - A\|_\infty$. Then it holds that*

$$\|\widehat{X} - \mathbf{u}_1\mathbf{u}_1^T\|_F \leq C(B)\frac{|\mathcal{S}|\lambda}{\lambda_r}, \tag{19}$$

*as long as $\lambda \geq C\lambda_1\sqrt{\frac{\log p}{n}}$.*

In Corollary 1, we only require $\lambda \geq C\lambda_1\sqrt{\frac{\log p}{n}}$ instead of $\lambda \geq C\lambda_1\sqrt{|\mathcal{S}|}\sqrt{\frac{\log p}{n}}$ (There is an extra $\sqrt{|\mathcal{S}|}$ in Theorem 7). This is because Theorem 7 considers the worst case when noise terms cannot be cancelled out.

The detailed estimation procedure of (P2) is provided in Section 15. Although the SDP method is stable and can find the global optimal very well, it suffers from high computational burden and expensive storage cost. The numerical comparison between the semidefinite programming methods and the proposed NC-SGEP method is given in the main context. It shows that our method is competitive compared with SDP-type methods in terms of estimation error.

## 8    Additional Numerical Experiments

We present the additional simulation results to show the non-convex regularized estimator enjoys merits of sparsity and stability under various settings.

**Sparse CCA**    Let $\Sigma_x$ and $\Sigma_y$ be two $p_1$ by $p_1$ matrices. Both of them take the form as
$$\begin{pmatrix} 2 & 1 & & & \\ 1 & 2 & 1 & & \\ & \ddots & \ddots & \ddots & \\ & & 1 & 2 & 1 \\ & & & 1 & 2 \end{pmatrix}.$$ We take $\Sigma_{xy} = 0.9\Sigma_x\mathbf{u}_1\mathbf{u}_1^T\Sigma_y$ and unit vector $\mathbf{u}_1$ only has two non-
zero entries. We set $n = 100$ and let $p = 2p_1$ vary from $50 - 400$. We compare the performance of proposed estimator with Truncated Rayleigh Flow Method (Rifle, [30]), truncated version of Generalized Eigenvector via Linear System Solver (GenELin, [11]). For fair comparison, we randomly choose the starting values. The estimation errors are reported in Table 3. Additionally, we also report the probability of successfully finding the global solution for each method in Table 3. The results show that our method has better performance in estimating the leading eigenvector. In addition, NC-SGEP is more easily to find the optimal solution and is thus more stable and accurate.

Table 3: Estimation accuracy for sparse canonical correlation analysis. "Err": $\|\hat{\mathbf{x}} - \mathbf{u}_1\|$ with standard error in parenthesis. "Opt": the percentage of finding global optimum. Each setting is replicated for 100 times.

| | $p$ | 100 | 200 | 300 | 400 |
|---|---|---|---|---|---|
| | **NC-SGEP** | 0.116 (0.127) | 0.137 (0.153) | 0.128 (0.118) | 0.241 (0.288) |
| Err | Rifle | 0.203 (0.121) | 0.368 (0.256) | 0.542 (0.278) | 0.662 (0.240) |
| | GenELin | 0.222 (0.127) | 0.485 (0.333) | 0.466 (0.325) | 0.719 (0.278) |
| | **NC-SGEP** | 96 % | 94 % | 93 % | 82 % |
| Opt | Rifle | 94 % | 67 % | 40 % | 24 % |
| | GenELin | 96 % | 60 % | 54 % | 30 % |

**Sparse FDA** Given two classes, the data from the first class follows $N(\mu_1, I_p)$ and the data from the second class follows $N(\mu_2, I_p)$. The mean vector $\mu_1$ satisfies $\mu_1[j] = 1$ for $j = 1, \ldots, 4$ and $\mu_1[j] = 0$ for $j > 4$; $\mu_2$ satisfies $\mu_2[j] = -1$ for $j = 1, \ldots, 4$ and $\mu_2[j] = 0$ for $j > 4$. We sample 100 data points from each class and split them into test sets and training sets into two equal half. Thus, $n = n_{train} = 100$ and $n_{test} = 100$. We let $p$ vary from 50 to 250 and let sparsity level vary from 10 to 50 for methods, NC-SGEP, Rifle and GenELin. Due to the penalization, our method does

Table 4: Classification accuracy for sparse Fisher's discriminant analysis problem.

| | $p$ | 50 | 100 | 150 | 200 | 250 |
|---|---|---|---|---|---|---|
| | **NC-SGEP** | 0.973 (0.013) | 0.976 (0.016) | 0.971 (0.016) | 0.974 (0.014) | 0.972 (0.018) |
| $s_n = 10$ | Rifle | 0.963 (0.019) | 0.959 (0.027) | 0.958 (0.020) | 0.954 (0.023) | 0.953 (0.020) |
| | GenELin | 0.962 (0.020) | 0.961 (0.241) | 0.959 (0.205) | 0.960 (0.020) | 0.959 (0.020) |
| | **NC-SGEP** | 0.969 (0.016) | 0.972 (0.019) | 0.970 (0.013) | 0.971 (0.019) | 0.975 (0.016) |
| $s_n = 20$ | Rifle | 0.942 (0.027) | 0.940 (0.030) | 0.930 (0.028) | 0.935 (0.029) | 0.932 (0.029) |
| | GenELin | 0.941 (0.026) | 0.937 (0.024) | 0.927 (0.024) | 0.928 (0.032) | 0.931 (0.026) |
| | **NC-SGEP** | 0.959 (0.024) | 0.946 (0.028) | 0.951 (0.027) | 0.954 (0.025) | 0.958 (0.027) |
| $s_n = 50$ | Rifle | 0.910 (0.036) | 0.867 (0.037) | 0.860 (0.048) | 0.869 (0.041) | 0.859 (0.037) |
| | GenELin | 0.907 (0.039) | 0.850 (0.034) | 0.839 (0.041) | 0.846 (0.047) | 0.848 (0.041) |

not over-select the features. Thus the proposed method is better and more stable compare with other existing methods in this task.

# 9 Proofs of Upper and Lower Perturbation Bounds

**Proof of Lemma 2** Let $U = [\hat{\mathbf{u}}_1, \mathbf{u}_2, \dots, \mathbf{u}_p]$, $\Lambda = \mathrm{diag}(\lambda_1, \lambda_2, \dots, \lambda_p)$ be such that

$$U^{\mathrm{T}}AU = \Lambda, \quad U^{\mathrm{T}}BU = I_p, \tag{20}$$

where $\hat{\mathbf{u}}_1 = \frac{\mathbf{u}_1}{\sqrt{\mathbf{u}_1^{\mathrm{T}}B\mathbf{u}_1}}$. Denote $U_1 = [\hat{\mathbf{u}}_1, \mathbf{u}_2]$, $U_2 = [\mathbf{u}_3, \dots, \mathbf{u}_p]$, $\Lambda_1 = \mathrm{diag}(\lambda_1, \lambda_2)$ and $\Lambda_2 = \mathrm{diag}(\lambda_3, \dots, \lambda_p)$.

Now let

$$E = BU_1 \begin{bmatrix} 0 & \epsilon \\ \epsilon & 0 \end{bmatrix} U_1^{\mathrm{T}}B, \qquad F = -E, \tag{21}$$

where $|\epsilon| \ll 1$ is a parameter. Let the eigenpairs of $\left( \begin{bmatrix} \lambda_1 & \epsilon \\ \epsilon & \lambda_2 \end{bmatrix}, \begin{bmatrix} 1 & -\epsilon \\ -\epsilon & 1 \end{bmatrix} \right)$ be $(\mu_1, \begin{bmatrix} 1 \\ \alpha \end{bmatrix})$, $(\mu_2, \begin{bmatrix} \beta \\ 1 \end{bmatrix})$, where $\mu_1 \geq \mu_2$, $\alpha, \beta \in \mathbb{R}$. Denote $\widetilde{U}_1 = [\tilde{\mathbf{u}}_1, \tilde{\mathbf{u}}_2] = U_1 \begin{bmatrix} 1 & \beta \\ \alpha & 1 \end{bmatrix}$. It follows that

$$\begin{aligned}
\widetilde{A}\widetilde{U}_1 &= (A + E)U_1 \begin{bmatrix} 1 & \beta \\ \alpha & 1 \end{bmatrix} = BU_1 \begin{bmatrix} \lambda_1 & \epsilon \\ \epsilon & \lambda_2 \end{bmatrix} \begin{bmatrix} 1 & \beta \\ \alpha & 1 \end{bmatrix} \\
&= BU_1 \begin{bmatrix} 1 & -\epsilon \\ -\epsilon & 1 \end{bmatrix} \begin{bmatrix} 1 & \beta \\ \alpha & 1 \end{bmatrix} \mathrm{diag}(\mu_1, \mu_2) \\
&= (B + F)U_1 \begin{bmatrix} 1 & \beta \\ \alpha & 1 \end{bmatrix} \mathrm{diag}(\mu_1, \mu_2) \\
&= \widetilde{B}\widetilde{U}_1 \mathrm{diag}(\mu_1, \mu_2)
\end{aligned}$$

and

$$\begin{aligned}
\widetilde{A}U_2 &= (A + E)U_2 = AU_2 \\
&= BU_2\Lambda_2 = (B + F)U_2\Lambda_2 = \widetilde{B}U_2\Lambda_2.
\end{aligned}$$

In other words, the eigenpairs of $(\widetilde{A}, \widetilde{B})$ are $(\mu_1, \tilde{\mathbf{u}}_1), (\mu_2, \tilde{\mathbf{u}}_2), (\lambda_3, \mathbf{u}_3), \dots, (\lambda_p, \mathbf{u}_p)$.

On one hand, using (21) and $\hat{\mathbf{u}}_1 = \frac{\mathbf{u}_1}{\sqrt{\mathbf{u}_1^{\mathrm{T}}B\mathbf{u}_1}}$, by calculations, we have [2]

$$\sqrt{\|E\mathbf{u}_1\|^2 + \|F\mathbf{u}_1\|^2} \leq \sqrt{2}\epsilon\|B\|. \tag{22}$$

On the other hand, using $\begin{bmatrix} 1 \\ \alpha \end{bmatrix}$ is the leading eigenvector of $\left( \begin{bmatrix} \lambda_1 & \epsilon \\ \epsilon & \lambda_2 \end{bmatrix}, \begin{bmatrix} 1 & -\epsilon \\ -\epsilon & 1 \end{bmatrix} \right)$, by calculations, we get

$$\alpha = \frac{\lambda_1 + 1}{\lambda_1 - \lambda_2}\epsilon + \mathcal{O}(\epsilon^3). \tag{23}$$

---

[2] For simplicity, we may use $\|\cdot\|$ instead of $\|\cdot\|_2$ in the proof.

Using (20), we have

$$\|\tilde{\mathbf{u}}_1\| \le \|U_1\|\sqrt{1+\alpha^2} \le \|B^{-\frac{1}{2}}\|\|B^{\frac{1}{2}}U_1\|\sqrt{1+\alpha^2} \le \|B^{-\frac{1}{2}}\|\sqrt{1+\alpha^2}. \tag{24}$$

Denote $U_3 = [\mathbf{u}_2, \ldots, \mathbf{u}_p]$ and $\Lambda_3 = \mathrm{diag}(\lambda_2, \ldots, \lambda_p)$. By calculations, we have

$$|\sin\theta(\mathbf{u}_1, \tilde{\mathbf{u}}_1)| \overset{(a)}{=} \frac{1}{\|\tilde{\mathbf{u}}_1\|}\|(U_3^{\mathrm{T}}B^2 U_3)^{-\frac{1}{2}}U_3^{\mathrm{T}}B\tilde{\mathbf{u}}_1\| \ge \frac{\|U_3^{\mathrm{T}}B\tilde{\mathbf{u}}_1\|}{\|\tilde{\mathbf{u}}_1\|\|B\|^{\frac{1}{2}}} \overset{(b)}{\ge} \frac{|\alpha|}{\|\tilde{\mathbf{u}}_1\|\|B\|^{\frac{1}{2}}} \overset{(c)}{\ge} \frac{|\alpha|}{\sqrt{(1+\alpha^2)\kappa}}, \tag{25}$$

where (a) uses (20) and (5), (b) uses (20), (c) uses (24).

Now using (22), (23) and (25), we get

$$|\sin\theta(\mathbf{u}_1, \tilde{\mathbf{u}}_1)| \ge \frac{1}{\sqrt{\kappa}}\frac{\lambda_1+1}{\lambda_1-\lambda_2}\epsilon + \mathcal{O}(\epsilon^2) \ge \frac{(\lambda_1+1)\cos\phi_1\cos\phi_2}{\sqrt{2\kappa}\|B\|}\frac{\sqrt{\|E\mathbf{u}_1\|^2+\|F\mathbf{u}_1\|^2}}{\sin(\phi_1-\phi_2)} + \mathcal{O}(\epsilon^2). \tag{26}$$

Simple calculations give rise to

$$(1+\lambda_1)\cos\phi_1 = \sqrt{2}\sin(\phi_1+\frac{\pi}{4}) \ge 1,$$

$$\frac{\cos\phi_2}{\sigma_{\min}(B)} = \frac{1}{\sigma_{\min}(B)\sqrt{1+\lambda_2^2}} \ge \frac{1}{\sigma_{\min}(B)\sqrt{1+\lambda_1^2}} \ge \frac{1}{\sqrt{\|A\|^2+\|B\|^2}}.$$

Substituting them into (26), we get

$$|\sin\theta(\mathbf{u}_1, \tilde{\mathbf{u}}_1)| \ge \frac{C_l\,\xi}{\sin(\phi_1-\phi_2)} + \mathcal{O}(\epsilon^2).$$

The conclusion follows immediately.

**Proof of Theorem 1** In this proof, we use $[\mathbf{x}]_\mathcal{K}$ to denote the subvector of $\mathbf{x}$ with entries in set $\mathcal{K}$ and $X_\mathcal{K}$ is the submatrix of $X$ with both row and column indices in $\mathcal{K}$. Let $\mathcal{K}$ be a superset of $\mathcal{S}$ with $|\mathcal{K}| = \ell \le s + k$, denote the eigenpairs of $(A_\mathcal{K}, B_\mathcal{K})$ and $(\widetilde{A}_\mathcal{K}, \widetilde{B}_\mathcal{K})$ by $(\mu_1, w_1), \ldots, (\mu_\ell, w_\ell)$, and $(\tilde{\mu}_1, \tilde{w}_1), \ldots, (\tilde{\mu}_\ell, \tilde{w}_\ell)$, respectively, and $\mu_1 \ge \cdots \ge \mu_\ell$, $\tilde{\mu}_1 \ge \cdots \ge \tilde{\mu}_\ell$. We consider case $= \emptyset$ and $\ne \emptyset$ in order.

**Case $= \emptyset$** In such case, $\mathcal{K} = \mathcal{J}$. Consider $(A_\mathcal{K}, B_\mathcal{K})$ and $(\widetilde{A}_\mathcal{K}, \widetilde{B}_\mathcal{K})$. Obviously, $[\mathbf{u}_1]_\mathcal{K}$ and $[\mathbf{x}_*]_\mathcal{K}$ are leading eigenvectors of $(A_\mathcal{K}, B_\mathcal{K})$ and $(\widetilde{A}_\mathcal{K}, \widetilde{B}_\mathcal{K})$, respectively. Without loss of generality, let $\|\mathbf{u}_1\| = 1$. By Assumption **A1)**, we have

$$\sqrt{\|E_\mathcal{K}[\mathbf{u}_1]_\mathcal{K}\|^2+\|F_\mathcal{K}[\mathbf{u}_1]_\mathcal{K}\|^2} \le \sqrt{\|E_\mathcal{K}\|^2+\|F_\mathcal{K}\|^2} \le \epsilon c_\mathcal{K}.$$

Then it follows that

$$\xi_\mathcal{K} = \frac{\sqrt{\|E_\mathcal{K}[\mathbf{u}_1]_\mathcal{K}\|^2+\|F_\mathcal{K}[\mathbf{u}_1]_\mathcal{K}\|^2}}{\tilde{c}_\mathcal{K}} \le \frac{\epsilon c_\mathcal{K}}{\tilde{c}_\mathcal{K}}. \tag{27}$$

By Lemma 1, we have

$$|\sin\theta([\mathbf{u}_1]_\mathcal{K}, [\mathbf{x}_*]_\mathcal{K})| \le \frac{C_{u,\mathcal{K}}\,\xi_\mathcal{K}}{\sin(\phi_1-\tilde{\phi}_2)}, \tag{28}$$

where $C_{u,\mathcal{K}} = \frac{\sqrt{2(\|A_\mathcal{K}\|^2+\|B_\mathcal{K}\|^2)}}{c_\mathcal{K}}$, $\tan\tilde{\phi}_2$ is the second largest eigenvalue of $(\widetilde{A}_\mathcal{K}, \widetilde{B}_\mathcal{K})$. Combining (27) and (28), we have

$$|\sin\theta(\mathbf{u}_1, \mathbf{x}_*)| = |\sin\theta([\mathbf{u}_1]_\mathcal{K}, [\mathbf{x}_*]_\mathcal{K})| \le \frac{\epsilon\sqrt{2(\|A_\mathcal{K}\|^2+\|B_\mathcal{K}\|^2)}}{\tilde{c}_\mathcal{K}\sin(\phi_1-\tilde{\phi}_2)}.$$

**Case $\ne \emptyset$** First, similar to **Case $= \emptyset$**, we have

$$|\sin\theta([\mathbf{u}_1]_\mathcal{K}, \tilde{w}_1)| \le \frac{\epsilon\sqrt{2(\|A_\mathcal{K}\|^2+\|B_\mathcal{K}\|^2)}}{\tilde{c}_\mathcal{K}\sin(\phi_1-\tilde{\phi}_2)}. \tag{29}$$

Second, without loss of generality, let $\|\mathbf{x}_*\| = 1$. Taking $[\mathbf{x}_*]_{\mathcal{K}}$ as an approximate eigenvector of $(\widetilde{A}_{\mathcal{K}}, \widetilde{B}_{\mathcal{K}})$, we are able to give upper bound for $|\sin\theta([\mathbf{x}_*]_{\mathcal{K}}, \tilde{w}_1)|$ as follows. Let

$$r = \widetilde{A}_{\mathcal{K}}[\mathbf{x}_*]_{\mathcal{K}} - \rho_* \widetilde{B}_{\mathcal{K}}[\mathbf{x}_*]_{\mathcal{K}}, \tag{30}$$

it is easy to see that

$$[\mathbf{x}_*]_{\mathcal{K}}^{\mathrm{T}} r = 0. \tag{31}$$

Now let

$$G = r[\mathbf{x}_*]_{\mathcal{K}}^{\mathrm{T}} + [\mathbf{x}_*]_{\mathcal{K}} r^{\mathrm{T}}, \qquad \widehat{E} = -\frac{G}{1+\rho_*^2}, \qquad \widehat{F} = \frac{\rho_* G}{1+\rho_*^2}. \tag{32}$$

Direct calculations give

$$(\widetilde{A}_{\mathcal{K}} + \widehat{E})[\mathbf{x}_*]_{\mathcal{K}} \overset{(a)}{=} \widetilde{A}_{\mathcal{K}}[\mathbf{x}_*]_{\mathcal{K}} - \frac{r}{1+\rho_*^2} \overset{(b)}{=} \rho_* \widetilde{B}_{\mathcal{K}}[\mathbf{x}_*]_{\mathcal{K}} + r - \frac{r}{1+\rho_*^2} = \rho_*(\widetilde{B}_{\mathcal{K}}[\mathbf{x}_*]_{\mathcal{K}} + \frac{\rho_* r}{1+\rho_*^2})$$

$$\overset{(c)}{=} \rho_*(\widetilde{B}_{\mathcal{K}} + \widehat{F})[\mathbf{x}_*]_{\mathcal{K}}, \tag{33}$$

where (a) and (c) use (31) and (32), (b) uses (30). In other words, $(\rho_*, [\mathbf{x}_*]_{\mathcal{K}})$ is an eigenpair of $(\widehat{A}_{\mathcal{K}}, \widehat{B}_{\mathcal{K}}) = (\widetilde{A}_{\mathcal{K}} + \widehat{E}, \widetilde{B}_{\mathcal{K}} + \widehat{F})$. Next, we show that $(\rho_*, [\mathbf{x}_*]_{\mathcal{K}})$ is the leading eigenpair.

Using (31), we know that $\left[\frac{r}{\|r\|}, [\mathbf{x}_*]_{\mathcal{K}}\right]$ is orthonormal, then it follows that

$$\|G\| = \|r\| \left\| \left[\frac{r}{\|r\|}, [\mathbf{x}_*]_{\mathcal{K}}\right] \left[[\mathbf{x}_*]_{\mathcal{K}}, \frac{r}{\|r\|}\right]^{\mathrm{T}} \right\| = \|r\|. \tag{34}$$

Using (30), by calculations, we have

$$\|r\| = \left\| \begin{bmatrix} \widetilde{A}_{\mathcal{J}}[\mathbf{x}_*]_{\mathcal{J}} - \rho_* \widetilde{B}_{\mathcal{J}}[\mathbf{x}_*]_{\mathcal{J}} \\ \widetilde{A}_{(,\mathcal{J})}[\mathbf{x}_*]_{\mathcal{J}} - \rho_* \widetilde{B}_{(,\mathcal{J})}[\mathbf{x}_*]_{\mathcal{J}} \end{bmatrix} \right\| = \left\| \begin{bmatrix} 0 \\ \widetilde{A}_{(,\mathcal{J})}[\mathbf{x}_*]_{\mathcal{J}} - \rho_* \widetilde{B}_{(,\mathcal{J})}[\mathbf{x}_*]_{\mathcal{J}} \end{bmatrix} \right\| = \delta \tilde{c}_{\mathcal{K}}. \tag{35}$$

Now using (32), (34) and (35), we get

$$\sqrt{\|\widehat{E}\|^2 + \|\widehat{F}\|^2} = \frac{\|G\|}{\sqrt{1+\rho_*^2}} = \frac{\delta \tilde{c}_{\mathcal{K}}}{\sqrt{1+\rho_*^2}}. \tag{36}$$

Since $\arctan\rho_* > \arctan\mu_2 + \arctan\epsilon + \arctan\frac{\delta}{\sqrt{1+\rho_*^2}}$, we have

$$\arctan\rho_* > \arctan\tilde{\mu}_2 + \arctan\frac{\delta}{\sqrt{1+\rho_*^2}} \geq \arctan\hat{\mu}_2. \tag{37}$$

i.e., $(\rho_*, [\mathbf{x}_*]_{\mathcal{K}})$ is the leading eigenpair of $(\widehat{A}_{\mathcal{K}}, \widehat{B}_{\mathcal{K}})$. Then using Lemma 1 and (36), we get

$$|\sin\theta([\mathbf{x}_*]_{\mathcal{K}}, \tilde{w}_1)| \leq \frac{\sqrt{2(\|\widehat{A}_{\mathcal{K}}\|^2 + \|\widehat{B}_{\mathcal{K}}\|^2)}\,\delta}{\hat{c}_{\mathcal{K}} \sin(\phi_* - \tilde{\phi}_2)\sqrt{1+\rho_*^2}}. \tag{38}$$

where $\hat{c}_{\mathcal{K}} = c(\widehat{A}_{\mathcal{K}}, \widehat{B}_{\mathcal{K}})$. Recall the following fact [25].

Let $(A, B)$, $(\widetilde{A}, \widetilde{B})$ be two symmetric matrix pairs, then it holds

$$|c(A, B) - c(\widetilde{A}, \widetilde{B})| \leq \sqrt{\|\widetilde{A} - A\|_2^2 + \|\widetilde{B} - B\|_2^2}.$$

Thus we have

$$\hat{c}_{\mathcal{K}} - \tilde{c}_{\mathcal{K}} \geq -\sqrt{\|\widehat{E}\|^2 + \|\widehat{F}\|^2} \overset{(a)}{=} -\frac{\delta \tilde{c}_{\mathcal{K}}}{\sqrt{1+\rho_*^2}},$$

where (a) uses (36). Substituting it into (38), we get

$$|\sin\theta([\mathbf{x}_*]_{\mathcal{K}}, \tilde{w}_1)| \leq \frac{\sqrt{2(\|\widehat{A}_{\mathcal{K}}\|^2 + \|\widehat{B}_{\mathcal{K}}\|^2)}\,\delta}{\tilde{c}_{\mathcal{K}}(\sqrt{1+\rho_*^2} - \delta) \sin(\phi_* - \tilde{\phi}_2)}. \tag{39}$$

Finally, by calculations, we get

$$|\sin\theta(\mathbf{u}_1, \mathbf{x}_*)| = |\sin\theta([\mathbf{u}_1]_{\mathcal{K}}, [\mathbf{x}_*]_{\mathcal{K}})|$$
$$\leq |\sin\theta([\mathbf{u}_1]_{\mathcal{K}}, \tilde{w}_1)| + |\sin\theta([\mathbf{x}_*]_{\mathcal{K}}, \tilde{w}_1)|,$$

$$\sqrt{2(\|\widehat{A}_{\mathcal{K}}\|^2 + \|\widehat{B}_{\mathcal{K}}\|^2)} \overset{(b)}{\leq} 2\sqrt{\|\widetilde{A}_{\mathcal{K}}\|^2 + \|\widetilde{B}_{\mathcal{K}}\|^2 + \frac{\delta^2 \tilde{c}_{\mathcal{K}}^2}{1 + \rho_*^2}},$$

where (b) uses (36). Combining them with (29) and (39), we arrive at the conclusion.

## 10  Characterization of Penalty Function

By a closer look at family $\mathcal{P}_\lambda$, it is not hard to see that any function $p_\lambda \in \mathcal{P}_\lambda$ satisfies the following properties:

1. $p_\lambda$ is locally equivalent to L1 norm around 0;
2. $p_\lambda$ is increasing and satisfies the uni-variate triangle inequality;
3. $p_\lambda(x)$ is a constant function when $x$ is large enough.

We first show the *local equivalence* between $p_\lambda$ and L1 norm.  For each $p_\lambda \in \mathcal{P}_\lambda$, it holds that there exist constants $c_1$, $c_2$ and $\delta_0 := c_0\lambda$ (constants may depend on $p_\lambda$) such that

$$c_1 p_\lambda^2(x) \leq \lambda^2 x^2 \quad \text{for all } x; \tag{40}$$
$$\lambda^2 x^2 \leq c_2 p_\lambda^2(x) \quad \text{for } |x| < \delta_0. \tag{41}$$

For particular penalty function, we can easily obtain $c_1$, $c_2$ and $\delta_0$:

**SCAD**: $c_1 = 1$; $c_2 = 1$ with $\delta_0 = \lambda$.
**MCP**: $c_1 = 1$; $c_2 = 4$ with $\delta_0 = a\lambda$.

**Proof of** (40) **and** (41).  First, (40) is obvious. This is because

$$p_\lambda(x) = \int_0^x p_\lambda^{'}(t)dt \leq \int_0^x \lambda dt = \lambda x$$

for any positive $x$. As a results, we can easily take $c_1 = 1$. Next we prove (41),

$$p_\lambda(x) = \int_0^x p_\lambda^{'}(t)dt \geq \int_0^x (\lambda - \kappa t)dt = \lambda x - \frac{\kappa}{2}x^2 \geq \lambda x - \frac{\lambda}{2}x = \frac{1}{2}\lambda x$$

for any $0 \leq x \leq \lambda/\kappa$. Thus, we can take $c_2 = 4$ and $\delta_0 = \lambda/\kappa$.

Next we show that $p_\lambda(x)$ satisfies *univariate triangle inequality*, i.e.,

$$p_\lambda(x + y) \leq p_\lambda(x) + p_\lambda(y) \tag{42}$$

holds for any $x, y \in \mathbb{R}$.

**Proof of** (42) First notice that $|x + y| \leq |x| + |y|$. Thus $p_\lambda(x + y) \leq p_\lambda(|x| + |y|)$. So it suffices to show that $p_\lambda(|x| + |y|) \leq p_\lambda(|x|) + p_\lambda(|y|) = p_\lambda(x) + p_\lambda(y)$.

By integration, we have that

$$p_\lambda(|x| + |y|) = \int_0^{|x|+|y|} p_\lambda^{'}(t)dt$$
$$= \int_0^{|x|} p_\lambda^{'}(t)dt + \int_{|x|}^{|x|+|y|} p_\lambda^{'}(t)dt$$
$$\leq \int_0^{|x|} p_\lambda^{'}(t)dt + \int_0^{|y|} p_\lambda^{'}(t)dt$$
$$= p_\lambda(|x|) + p_\lambda(|y|)$$
$$= p_\lambda(x) + p_\lambda(y),$$

where we use the monotonicity of $p_\lambda^{'}(x)$ and symmetry of $p_\lambda(x)$.

## 11 Proof for Estimation Bounds

**Difference between eigenvalues.** We first give the bound between $\widehat{\lambda}_i$ and $\lambda_i$. By Theorem 3.2 in [25], we know that

$$|\tan^{-1}(\widehat{\lambda}_i) - \tan^{-1}(\lambda_i)| \leq \sin^{-1}(\frac{\epsilon}{c(A,B)}), \tag{43}$$

with $\epsilon = \sqrt{\|E\|_2^2 + \|F\|_2^2}$. Thus, we get

$$\widehat{\lambda}_i - \lambda_i = (1 + \tan^2(\widetilde{\theta}_i)) \sin^{-1}(\frac{\epsilon}{c(A,B)}), \tag{44}$$

where $\widetilde{\theta}_i$ is some angle between $\tan^{-1}(\widehat{\lambda}_i)$ and $\tan^{-1}(\lambda_i)$.

**Difference between eigenvectors.** The bound of $\sin\theta(\widehat{\mathbf{u}}_i, \mathbf{u}_i)$ can be obtained from Theorem 2.1 in [28]. We know that

$$|\sin\theta(\widehat{\mathbf{u}}_i, \mathbf{u}_i)| \leq \frac{\sqrt{2}\sqrt{\|A^2 + B^2\|_2}}{c(A,B)c(\widetilde{A}, \widetilde{B})} \frac{\sqrt{\|E\mathbf{u}_i\|_2^2 + \|F\mathbf{u}_i\|_2^2}}{\delta_i}, \tag{45}$$

where $\delta_i$ is the $\min_{j \neq i}\{d_c(\lambda_i, \lambda_j)\}$ with chord distance $d_c(x,y)$ defined as $\frac{|x-y|}{\sqrt{(1+x^2)(1+y^2)}}$.

Therefore, the bound between $\widetilde{\mathbf{u}}_1$ and $\mathbf{u}_1$:

$$\begin{aligned}
\|\widetilde{\mathbf{u}}_1 - \mathbf{u}_1\|_2^2 &= (\widetilde{\mathbf{u}}_1 - \mathbf{u}_1)^{\mathrm{T}}(\widetilde{\mathbf{u}}_1 - \mathbf{u}_1) \\
&= 2 - \cos\theta(\widetilde{\mathbf{u}}_1, \mathbf{u}_1) \\
&\leq 2 - \cos^2\theta(\widetilde{\mathbf{u}}_1, \mathbf{u}_1) \leq 2\sin^2\theta(\widetilde{\mathbf{u}}_1, \mathbf{u}_1).
\end{aligned} \tag{46}$$

**Difference between the re-scaled eigenvectors.** We recall the re-scaled vector $\mathbf{u}_{1s} = \mathbf{u}_1(\mathbf{u}_1^{\mathrm{T}}B\mathbf{u}_1)^{-1/2}$ and define the corresponding perturbed version $\check{\mathbf{u}}_1 = \widetilde{\mathbf{u}}_1(\widetilde{\mathbf{u}}_1^{\mathrm{T}}\widetilde{B}\widetilde{\mathbf{u}}_1)^{-1/2}$ We can bound the difference between $\mathbf{u}_{1s}$ and $\check{\mathbf{u}}_1$ as follows.

$$\begin{aligned}
&\|\mathbf{u}_{1s} - \check{\mathbf{u}}_1\|_2 \\
=\ & \|\mathbf{u}_1(\mathbf{u}_1^{\mathrm{T}}B\mathbf{u}_1)^{-1/2} - \widetilde{\mathbf{u}}_1(\widetilde{\mathbf{u}}_1\widetilde{B}\widetilde{\mathbf{u}}_1)^{-1/2}\|_2 \\
=\ & (\mathbf{u}_1^{\mathrm{T}}B\mathbf{u}_1)^{-1/2}(\widetilde{\mathbf{u}}_1\widetilde{B}\widetilde{\mathbf{u}}_1)^{-1/2}\|\mathbf{u}_1(\widetilde{\mathbf{u}}_1\widetilde{B}\widetilde{\mathbf{u}}_1)^{1/2} - \widetilde{\mathbf{u}}_1(\mathbf{u}_1^{\mathrm{T}}B\mathbf{u}_1)^{1/2}\|_2 \\
\leq\ & (\mathbf{u}_1^{\mathrm{T}}B\mathbf{u}_1)^{-1/2}(\widetilde{\mathbf{u}}_1\widetilde{B}\widetilde{\mathbf{u}}_1)^{-1/2}(\|\mathbf{u}_1 - \widetilde{\mathbf{u}}_1\|_2(\mathbf{u}_1^{\mathrm{T}}B\mathbf{u}_1)^{1/2} \\
& + \frac{|\widetilde{\mathbf{u}}_1^{\mathrm{T}}\widetilde{B}\widetilde{\mathbf{u}}_1 - \mathbf{u}_1^{\mathrm{T}}\widetilde{B}\mathbf{u}_1|}{(\widetilde{\mathbf{u}}_1^{\mathrm{T}}\widetilde{B}\widetilde{\mathbf{u}}_1)^{1/2} + (\mathbf{u}_1^{\mathrm{T}}\widetilde{B}\mathbf{u}_1)^{1/2}} + \frac{|\mathbf{u}_1^{\mathrm{T}}\widetilde{B}\mathbf{u}_1 - \mathbf{u}_1^{\mathrm{T}}B\mathbf{u}_1|}{(\mathbf{u}_1^{\mathrm{T}}\widetilde{B}\mathbf{u}_1)^{1/2} + (\mathbf{u}_1^{\mathrm{T}}B\mathbf{u}_1)^{1/2}}) \\
\leq\ & (\mathbf{u}_1^{\mathrm{T}}B\mathbf{u}_1)^{-1/2}(\widetilde{\mathbf{u}}_1\widetilde{B}\widetilde{\mathbf{u}}_1)^{-1/2}(\|\mathbf{u}_1 - \widetilde{\mathbf{u}}_1\|_2(\mathbf{u}_1^{\mathrm{T}}B\mathbf{u}_1)^{1/2} \\
& + \frac{2\|\widetilde{B}\|_2\|\widetilde{\mathbf{u}}_1 - \mathbf{u}_1\|_2}{(\widetilde{\mathbf{u}}_1^{\mathrm{T}}\widetilde{B}\widetilde{\mathbf{u}}_1)^{1/2} + (\mathbf{u}_1^{\mathrm{T}}\widetilde{B}\mathbf{u}_1)^{1/2}} + \frac{\|\widetilde{B} - B\|_2}{(\mathbf{u}_1^{\mathrm{T}}\widetilde{B}\mathbf{u}_1)^{1/2} + (\mathbf{u}_1^{\mathrm{T}}B\mathbf{u}_1)^{1/2}}) \\
\leq\ & C_1(B)\|\widetilde{\mathbf{u}}_1 - \mathbf{u}_1\|_2 + C_2(B)\|\widetilde{B} - B\|_2, \tag{48}
\end{aligned}$$

where

$$C_1(B) = (\mathbf{u}_1^{\mathrm{T}}B\mathbf{u}_1)^{-1/2}(\widetilde{\mathbf{u}}_1\widetilde{B}\widetilde{\mathbf{u}}_1)^{-1/2}((\mathbf{u}_1^{\mathrm{T}}B\mathbf{u}_1)^{1/2} + \frac{2\|\widetilde{B}\|_2}{(\widetilde{\mathbf{u}}_1^{\mathrm{T}}\widetilde{B}\widetilde{\mathbf{u}}_1)^{1/2} + (\mathbf{u}_1^{\mathrm{T}}\widetilde{B}\mathbf{u}_1)^{1/2}})$$

and

$$C_2(B) = (\mathbf{u}_1^{\mathrm{T}}B\mathbf{u}_1)^{-1/2}(\widetilde{\mathbf{u}}_1\widetilde{B}\widetilde{\mathbf{u}}_1)^{-1/2}/((\mathbf{u}_1^{\mathrm{T}}\widetilde{B}\mathbf{u}_1)^{1/2} + (\mathbf{u}_1^{\mathrm{T}}B\mathbf{u}_1)^{1/2}).$$

Remark: $C_1(B) \equiv 1$ if $\widetilde{B} = B = I$.

**Characterization of $\hat{\mathbf{u}}_1$** By recalling the definition of $\hat{\mathbf{u}}_1$ that

$$\hat{\mathbf{u}}_1 = \arg\max_{\mathbf{x} \in \mathbb{R}^p, \|\mathbf{x}\|_0 \leq s_n} \mathbf{x}^{\mathrm{T}}\widetilde{A}\mathbf{x}; \quad \text{subject to } \mathbf{x}^{\mathrm{T}}\widetilde{B}\mathbf{x} = 1. \tag{49}$$

Next, we show that $\mathcal{S} \subset \operatorname{supp}(\hat{\mathbf{u}}_1)$. We prove this via using contradiction method.

Denote $\tilde{\rho} := \hat{\mathbf{u}}_1 \tilde{A} \hat{\mathbf{u}}_1$ and denote $\tilde{\rho}_{\mathcal{S}} := \max_{\mathbf{x} \in \mathbb{R}^p, \operatorname{supp}(\mathbf{x}) = \mathcal{S}, \mathbf{x}^{\mathrm{T}} \tilde{B} \mathbf{x} = 1} \mathbf{x}^{\mathrm{T}} \tilde{A} \mathbf{x}$. By the definition of $\hat{\mathbf{u}}_1$, we have $\tilde{\rho} \geq \tilde{\rho}_{\mathcal{S}}$. Denote $\mathcal{S}_u := \operatorname{supp}(\hat{\mathbf{u}}_1)$. Then by the perturbation results, we have that

$$\tilde{\rho} \leq \lambda_1(\mathcal{S}_u) + O(\|E[\mathcal{S}_u, \mathcal{S}_u]\|_2 + F[\mathcal{S}_u, \mathcal{S}_u]\|_2), \tag{50}$$

where $\lambda_1(\mathcal{S}_u)$ is the leading eigenvalue of matrix pair $(A[\mathcal{S}_u, \mathcal{S}_u], B[\mathcal{S}_u, \mathcal{S}_u])$. We also have

$$\tilde{\rho}_{\mathcal{S}} \geq \lambda_1 + O(\|E[\mathcal{S}, \mathcal{S}]\|_2 + F[\mathcal{S}, \mathcal{S}]\|_2). \tag{51}$$

By Condition that $\lambda_1 > \lambda(\mathcal{S}_u)$ when $\mathcal{S} \not\subset \mathcal{S}_u$ and the fact $\|E[\mathcal{S}, \mathcal{S}]\|_2 + F[\mathcal{S}, \mathcal{S}]\|_2 + \|E[\mathcal{S}_u, \mathcal{S}_u]\|_2 + F[\mathcal{S}_u, \mathcal{S}_u]\|_2 \ll 1$, thus $\tilde{\rho} < \tilde{\rho}_{\mathcal{S}}$. This contradicts with the definition of $\hat{\mathbf{u}}_1$. Hence we conclude that $\mathcal{S} \subset \operatorname{supp}(\hat{\mathbf{u}}_1)$.

**Proof of Theorem 3** Similar to the notations in the previous section, we define the truncated vector $\hat{\mathbf{u}}_K$ such that $\hat{\mathbf{u}}_K[\mathcal{S}] = \hat{\mathbf{u}}_1[\mathcal{S}]$ and $\hat{\mathbf{u}}_K[-\mathcal{S}] = \mathbf{0}$ and let $\hat{\mathbf{u}}_{-K} = \hat{\mathbf{u}}_1 - \hat{\mathbf{u}}_K$. We also define some constants $b_1 := \max_{\mathcal{K}: \|\mathcal{K}\|_0 \leq s_n} \|B_{\mathcal{K}}\|_2$ and $b_2 := \max_{\mathcal{K}: \|\mathcal{K}\|_0 \leq s_n} \|(B_{\mathcal{K}})^{-1}\|_2$ which are related to underlying matrix $B$.

We first construct an auxiliary vector $\hat{\mathbf{u}}_r$ such that

$$\hat{\mathbf{u}}_r := \alpha \hat{\mathbf{u}}_K, \ \alpha = \sqrt{\frac{\hat{\mathbf{u}}_K^{\mathrm{T}} \tilde{B} \hat{\mathbf{u}}_K + 2 \hat{\mathbf{u}}_K^{\mathrm{T}} \tilde{B} \hat{\mathbf{u}}_{-K}}{\hat{\mathbf{u}}_K^{\mathrm{T}} \tilde{B} \hat{\mathbf{u}}_K}}. \tag{52}$$

We can similarly verify that

$$\begin{aligned}
\hat{\mathbf{u}}_r^{\mathrm{T}} \tilde{B} \hat{\mathbf{u}}_r &= \hat{\mathbf{u}}_K^{\mathrm{T}} \tilde{B} \hat{\mathbf{u}}_K + 2 \hat{\mathbf{u}}_K^{\mathrm{T}} \tilde{B} \hat{\mathbf{u}}_{-K} \\
&\leq \hat{\mathbf{u}}_1^{\mathrm{T}} \tilde{B} \hat{\mathbf{u}}_1 \leq 1.
\end{aligned}$$

We also know that

$$\|\hat{\mathbf{u}}_{-K}\|_2 = \|\hat{\mathbf{u}}_1[-\mathcal{S}_u]\|_2 \leq \sqrt{2} b_2^{1/2} \sin \theta(\hat{\mathbf{u}}_1, \mathbf{u}_1).$$

Additionally, we can also compute that

$$\begin{aligned}
\hat{\mathbf{u}}_r^{\mathrm{T}} \tilde{A} \hat{\mathbf{u}}_r &= \hat{\mathbf{u}}_1 \tilde{A} \hat{\mathbf{u}}_1 - \tilde{\epsilon}^{\mathrm{T}} \hat{\mathbf{u}}_{-K} \\
&= \tilde{\rho} - \tilde{\epsilon}^{\mathrm{T}} \hat{\mathbf{u}}_{-K},
\end{aligned} \tag{53}$$

where $\tilde{\epsilon} := \tilde{A} \hat{\mathbf{u}}_{-K} + \tilde{\epsilon}_{AB} - 2 \frac{\tilde{\epsilon}_{AB}^{\mathrm{T}}}{\hat{\mathbf{u}}_K^{\mathrm{T}} \tilde{B} \hat{\mathbf{u}}_K} \tilde{B} \hat{\mathbf{u}}_K$ and $\tilde{\epsilon}_{AB} = \tilde{A} \hat{\mathbf{u}}_K - \tilde{\rho} \tilde{B} \hat{\mathbf{u}}_K$.

For $\tilde{\epsilon}_{AB}$, we know that

$$\begin{aligned}
\|\tilde{\epsilon}_{AB}[\mathcal{S}_u]\|_2 &= \|\tilde{A}[\mathcal{S}_u, :] \hat{\mathbf{u}}_K - \tilde{\rho} \tilde{B}[\mathcal{S}_u, :] \hat{\mathbf{u}}_K\|_2 \\
&= \|\tilde{A}[\mathcal{S}_u, :] \hat{\mathbf{u}}_K - \tilde{A}[\mathcal{S}_u, :] \hat{\mathbf{u}}_1\|_2 + \|\tilde{\rho} \tilde{B}[\mathcal{S}_u, :] \hat{\mathbf{u}}_1 - \tilde{\rho} \tilde{B}[\mathcal{S}_u, :] \hat{\mathbf{u}}_K\|_2 \\
&\leq (\|\tilde{A}_{\mathcal{S}_u}\|_2 + \tilde{\rho} \|\tilde{B}_{\mathcal{S}_u}\|_2) \|\hat{\mathbf{u}}_K - \tilde{\mathbf{u}}_1\|_2 \\
&\leq 2 \tilde{\rho} b_1 b_2^{1/2} \|\tilde{\mathbf{u}}_1 - \mathbf{u}_1\|_2.
\end{aligned}$$

Therefore,

$$\begin{aligned}
\|\tilde{\epsilon}[\mathcal{S}_u]\|_2 &= \|\tilde{A}[\mathcal{S}_u, :] \hat{\mathbf{u}}_{-K} + \tilde{\epsilon}_{AB}[\mathcal{S}_u] - 2 \frac{\tilde{\epsilon}_{AB}^{\mathrm{T}} \hat{\mathbf{u}}_K[\mathcal{S}_u]}{\hat{\mathbf{u}}_K^{\mathrm{T}} \tilde{B} \hat{\mathbf{u}}_K} \tilde{B} \hat{\mathbf{u}}_K\|_2 \\
&\leq \|\tilde{A}_{\mathcal{S}_u}\|_2 \|\hat{\mathbf{u}}_{-K}\|_2 + \|\tilde{\epsilon}_{AB}[\mathcal{S}_u]\|_2 + 2(b_1 b_2)^{1/2} \|\tilde{\epsilon}_{AB}[\mathcal{S}_u]\|_2 \\
&\leq (1 + 2(1 + 2(b_1 b_2)^{1/2})) \tilde{\rho} b_1 b_2^{1/2} \|\tilde{\mathbf{u}}_1 - \mathbf{u}_1\|_2 \\
&:= C_1'(b_1, b_2) \|\tilde{\mathbf{u}}_1 - \mathbf{u}_1\|_2, 
\end{aligned} \tag{54}$$

which is $O(\epsilon)$. Therefore, together with (53), we have

$$\begin{aligned}
\hat{\mathbf{u}}_r^{\mathrm{T}} \tilde{A} \hat{\mathbf{u}}_r &= \tilde{\rho} - \tilde{\epsilon}^{\mathrm{T}} \hat{\mathbf{u}}_{-K} \\
&\geq \tilde{\rho} - C_1'(b_1, b_2) b_2^{1/2} \|\tilde{\mathbf{u}}_1 - \mathbf{u}_1\|_2^2 \\
&\geq \tilde{\rho} - C_3'(b_1, b_2) \|\tilde{\mathbf{u}}_1 - \mathbf{u}_1\|_2^2
\end{aligned} \tag{55}$$

with $C_3'(b_1, b_2) := (1 + 2(1 + 2(b_1 b_2)^{1/2}))\tilde{\rho} b_1 b_2$.

We know that $\hat{\mathbf{x}}$ can be represented as $\tilde{\alpha}_1 \hat{\mathbf{u}}_1 + \ldots + \tilde{\alpha}_{s_n} \hat{\mathbf{u}}_{s_n}$, where $[\hat{\mathbf{u}}_1, \ldots, \hat{\mathbf{u}}_{s_n}] = [\tilde{\mathbf{u}}_1, \ldots, \tilde{\mathbf{u}}_{s_n}] \cdot (\tilde{U}^{\mathrm{T}} \tilde{B} \tilde{U})^{-1/2}$ are the eigenvectors for $(\tilde{A}_{\mathcal{S}_u}, \tilde{B}_{\mathcal{S}_u})$ corresponding to eigenvalues $\tilde{\lambda}_1, \ldots, \tilde{\lambda}_{s_n}$. By the optimality of $\hat{\mathbf{x}}$ and the fact that $\hat{\mathbf{u}}_r^{\mathrm{T}} \tilde{B} \hat{\mathbf{u}}_r \leq 1$, it must hold that

$$-\hat{\mathbf{x}}^{\mathrm{T}} \tilde{A} \hat{\mathbf{x}} + p_\lambda(\hat{\mathbf{x}}) \leq -\hat{\mathbf{u}}_r^{\mathrm{T}} \tilde{A} \hat{\mathbf{u}}_r + p_\lambda(\hat{\mathbf{u}}_r). \tag{56}$$

This gives us that

$$\begin{aligned}
-\tilde{\alpha}_1^2 \tilde{\lambda}_1 - \ldots - \tilde{\alpha}_{s_n}^2 \tilde{\lambda}_{s_n} &\leq -\hat{\mathbf{u}}_r^{\mathrm{T}} \tilde{A} \hat{\mathbf{u}}_r + p_\lambda(\tilde{\mathbf{u}}_r) \\
(1 - \tilde{\alpha}_1^2)(\tilde{\lambda}_1 - \tilde{\lambda}_2) &\leq \|\tilde{\epsilon}^{\mathrm{T}} \hat{\mathbf{u}}_{-K}\|_2 + p_\lambda(\hat{\mathbf{u}}_r),
\end{aligned}$$

where we use the fact that $\tilde{\lambda}_1 = \rho$.

Notice that $\tilde{\lambda}_1 \geq \lambda_1 - \epsilon$ and $\tilde{\lambda}_2 \leq \lambda_2(A_{\mathcal{S}_u}, B_{\mathcal{S}_u}) + \epsilon \leq \lambda_2 + \epsilon$. It leads to $\tilde{\lambda}_1 - \tilde{\lambda}_2 \geq (\lambda_1 - \lambda_2)/2$. Therefore, $\tilde{\alpha}_1^2 \geq 1 - \frac{2}{\lambda_1 - \lambda_2}\{\|\tilde{\epsilon}^{\mathrm{T}} \hat{\mathbf{u}}_{-K}\|_2 + p_\lambda(\hat{\mathbf{u}}_r)\}$. In other words, $\tilde{\alpha}_1^2 \geq 1 - O(|\mathcal{S}|\lambda^2)$. This implies that

$$\begin{aligned}
&\|\hat{\mathbf{x}} - \mathbf{u}_{1s}\|_2 \\
\leq\ &\|\hat{\mathbf{x}} - \hat{\mathbf{u}}_1\|_2 + \|\hat{\mathbf{u}}_1 - \mathbf{u}_{1s}\|_2 \\
\leq\ &b_2^{1/2}(\sqrt{(1 - \tilde{\alpha}_1)^2 + \tilde{\alpha}_2^2 + \ldots, \tilde{\alpha}_p^2} + \|\hat{\mathbf{u}}_1 - \mathbf{u}_{1s}\|_2) \\
\leq\ &b_2^{1/2}(\sqrt{2(1 - \tilde{\alpha}_1^2)/(1 + \tilde{\alpha}_1)} + \|\hat{\mathbf{u}}_1 - \mathbf{u}_{1s}\|_2) \\
=\ &O(\sqrt{|\mathcal{S}|}\lambda). 
\end{aligned} \tag{57}$$

Thus the estimator is consistent.

By the condition that $\min\{\mathbf{u}_1[j] : j \in \mathcal{S}\} \gg \sqrt{|\mathcal{S}|}\lambda$, it implies that

$$p_\lambda(\hat{x}) = p_\lambda(\mathbf{u}_{1s}) = |\mathcal{S}| p_\lambda(\gamma\lambda), \tag{58}$$

since $\mathbf{u}_{1s}[\mathcal{S}] \gg \sqrt{|\mathcal{S}|}\lambda$ element-wisely. In addition, we observe that $p_\lambda(\tilde{\mathbf{u}}_r)$ is no greater than $|\mathcal{S}| p_\lambda(\gamma\lambda)$ since $\tilde{\mathbf{u}}_r$ has at most $|\mathcal{S}|$ non-zero elements. By these facts, we can improve the error bound of $\|\hat{\mathbf{x}} - \mathbf{u}_{1r}\|_2$. Again, by the definition of $\hat{\mathbf{x}}$, we have

$$-\hat{\mathbf{x}}^{\mathrm{T}} \tilde{A} \hat{\mathbf{x}} + p_\lambda(\hat{\mathbf{x}}) \leq -\tilde{\mathbf{u}}_r^{\mathrm{T}} \tilde{A} \tilde{\mathbf{u}}_r + p_\lambda(\tilde{\mathbf{u}}_r). \tag{59}$$

It gives us that

$$\begin{aligned}
-\tilde{\alpha}_1^2 \tilde{\lambda}_1 - \ldots - \tilde{\alpha}_{s_n}^2 \tilde{\lambda}_{s_n} &\leq -\hat{\mathbf{u}}_r^{\mathrm{T}} \tilde{A} \hat{\mathbf{u}}_r \\
(1 - \tilde{\alpha}_1^2)(\tilde{\lambda}_1 - \tilde{\lambda}_2) &\leq \|\tilde{\epsilon}^{\mathrm{T}} \hat{\mathbf{u}}_{-K}\|_2.
\end{aligned}$$

Therefore, $\tilde{\alpha}_1^2 \geq 1 - \frac{1}{\tilde{\lambda}_1 - \tilde{\lambda}_2}\|\tilde{\epsilon}^{\mathrm{T}} \hat{\mathbf{u}}_{-K}\|_2$. This implies that

$$\begin{aligned}
&\|\hat{\mathbf{x}} - \mathbf{u}_{1s}\|_2 \\
\leq\ &\|\hat{\mathbf{x}} - \hat{\mathbf{u}}_1\|_2 + \|\hat{\mathbf{u}}_1 - \mathbf{u}_{1s}\|_2 \\
\leq\ &b_2^{1/2}\sqrt{(1 - \tilde{\alpha}_1)^2 + \tilde{\alpha}_2^2 + \ldots, \tilde{\alpha}_p^2} + \|\hat{\mathbf{u}}_1 - \mathbf{u}_{1s}\|_2 \\
=\ &b_2^{1/2}\sqrt{\frac{1}{\tilde{\lambda}_1 - \tilde{\lambda}_2}}\sqrt{\|\tilde{\epsilon}^{\mathrm{T}} \hat{\mathbf{u}}_{-K}\|_2} + \|\hat{\mathbf{u}}_1 - \mathbf{u}_{1s}\|_2 \\
\leq\ &C(\sqrt{(1 + 2(1 + 2(b_1 b_2)^{1/2}))\tilde{\rho} b_1 b_2/(\lambda_1 - \lambda_2)} + b_1 b_2 + (b_1 b_2)^{1/2})b_2^{1/2}\|\tilde{\mathbf{u}}_1 - \mathbf{u}_1\|_2 + b_2^{3/2}\|\tilde{B} - B\|_2,
\end{aligned}$$

where the last inequality uses (48) by treating $\hat{\mathbf{u}}_1$ as $\check{\mathbf{u}}_1$. (This is valid since $\mathrm{supp}(\mathbf{u}_1) \subset \mathrm{supp}(\hat{\mathbf{u}}_1)$ and $\hat{\mathbf{u}}_1^{\mathrm{T}} \tilde{B} \hat{\mathbf{u}}_1 = 1$.) This concludes the error bound.

**Proof of Theorem 4** Next, we show that $\widehat{\mathbf{x}}$ satisfies the oracle property. We abuse the notation of $\widehat{\mathbf{u}}_K$ and $\mathbf{u}_{-K}$ by letting $\widehat{\mathbf{u}}_K$ be the vector such that $\widehat{\mathbf{u}}_K[\mathcal{S}] = \widehat{\mathbf{x}}[\mathcal{S}]$ and $\widehat{\mathbf{u}}_K[\mathcal{S}] = \mathbf{0}$, where $\mathcal{S}$ is the support of $\mathbf{u}_1$. Let $\widehat{\mathbf{u}}_{-K} = \widehat{\mathbf{x}} - \widehat{\mathbf{u}}_K$. Define $\widehat{\epsilon}_{AB} := \widetilde{A}\widehat{\mathbf{u}}_K - \widetilde{\lambda}_1\widetilde{B}\widehat{\mathbf{u}}_K$.

Consider a candidate solution $\widehat{\mathbf{u}}_0 = \alpha\widehat{\mathbf{u}}_K$, where coefficient $\alpha$ will be determined later. We aim to find a $\widehat{\mathbf{u}}_0$ such that

$$\widehat{\mathbf{u}}_0^{\mathrm{T}}\widetilde{A}\widehat{\mathbf{u}}_0 \;\geq\; \widehat{\mathbf{x}}^{\mathrm{T}}\widetilde{A}\widehat{\mathbf{x}} - \|\widehat{\mathbf{u}}_{-K}\|_2 \cdot O(\epsilon) \tag{60}$$

and

$$\widehat{\mathbf{u}}_0^{\mathrm{T}}\widetilde{A}\widehat{\mathbf{u}}_0 \;\leq\; 1. \tag{61}$$

We consider to take $\alpha^2 = 1 + \frac{2\widehat{\mathbf{u}}_K^{\mathrm{T}}\widetilde{B}\widehat{\mathbf{u}}_{-K}}{\widehat{\mathbf{u}}_K^{\mathrm{T}}\widetilde{B}\widehat{\mathbf{u}}_K}$. Then we can compute that

$$\begin{aligned}
\widehat{\mathbf{u}}_0^{\mathrm{T}}\widetilde{B}\widehat{\mathbf{u}}_0 &= \alpha^2\widehat{\mathbf{u}}_K^{\mathrm{T}}\widetilde{B}\widehat{\mathbf{u}}_K \\
&= (1 + \frac{2\widehat{\mathbf{u}}_K^{\mathrm{T}}\widetilde{B}\widehat{\mathbf{u}}_{-K}}{\widehat{\mathbf{u}}_K^{\mathrm{T}}\widetilde{B}\widehat{\mathbf{u}}_K})\widehat{\mathbf{u}}_K^{\mathrm{T}}\widetilde{B}\widehat{\mathbf{u}}_K \\
&= \widehat{\mathbf{u}}_K^{\mathrm{T}}\widetilde{B}\widehat{\mathbf{u}}_K + 2\widehat{\mathbf{u}}_K^{\mathrm{T}}\widetilde{B}\widehat{\mathbf{u}}_{-K} \\
&\leq \widehat{\mathbf{x}}^{\mathrm{T}}\widetilde{B}\widehat{\mathbf{x}} \leq 1.
\end{aligned}$$

We can also compute that

$$\begin{aligned}
\widehat{\mathbf{u}}_0^{\mathrm{T}}\widetilde{A}\widehat{\mathbf{u}}_0 &= \alpha^2\widehat{\mathbf{u}}_K^{\mathrm{T}}\widetilde{A}\widehat{\mathbf{u}}_K \\
&= \alpha^2(\widetilde{\lambda}\widetilde{B}\widehat{\mathbf{u}}_K + \widehat{\epsilon}_{AB})^{\mathrm{T}}\widehat{\mathbf{u}}_K \\
&= \alpha^2\widetilde{\lambda}\widehat{\mathbf{u}}_K^{\mathrm{T}}\widetilde{B}\widehat{\mathbf{u}}_K + \alpha^2\widehat{\epsilon}_{AB}^{\mathrm{T}}\widehat{\mathbf{u}}_K \\
&= \widetilde{\lambda}\widehat{\mathbf{u}}_K^{\mathrm{T}}\widetilde{B}\widehat{\mathbf{u}}_K + \widetilde{\lambda}\widehat{\mathbf{u}}_K^{\mathrm{T}}\widetilde{B}\widehat{\mathbf{u}}_{-K} + \alpha^2\widehat{\epsilon}_{AB}^{\mathrm{T}}\widehat{\mathbf{u}}_K \\
&= (\widetilde{\lambda}\widehat{\mathbf{u}}_K^{\mathrm{T}}\widetilde{B} + \widehat{\epsilon}_{AB}^{\mathrm{T}})\widehat{\mathbf{u}}_K + (\widetilde{\lambda}\widehat{\mathbf{u}}_K^{\mathrm{T}}\widetilde{B} + \widehat{\epsilon}_{AB}^{\mathrm{T}})\widehat{\mathbf{u}}_{-K} \\
&\quad -\widehat{\epsilon}_{AB}^{\mathrm{T}}\widehat{\mathbf{u}}_K - \widehat{\epsilon}_{AB}^{\mathrm{T}}\widehat{\mathbf{u}}_{-K} + \alpha^2\widehat{\epsilon}_{AB}^{\mathrm{T}}\widehat{\mathbf{u}}_K \\
&= \widehat{\mathbf{x}}^{\mathrm{T}}\widetilde{A}\widehat{\mathbf{x}} - \widehat{\mathbf{u}}_{-K}^{\mathrm{T}}\widetilde{A}\widehat{\mathbf{u}}_{-K} - \widehat{\epsilon}_{AB}^{\mathrm{T}}\widehat{\mathbf{u}}_K - \widehat{\epsilon}_{AB}^{\mathrm{T}}\widehat{\mathbf{u}}_{-K} + \alpha^2\widehat{\epsilon}_{AB}^{\mathrm{T}}\widehat{\mathbf{u}}_K \\
&= \widehat{\mathbf{x}}^{\mathrm{T}}\widetilde{A}\widehat{\mathbf{x}} - \widehat{\mathbf{u}}_{-K}^{\mathrm{T}}\widetilde{A}\widehat{\mathbf{u}}_{-K} - \epsilon_{AB}^{\mathrm{T}}\widehat{\mathbf{u}}_{-K} + \frac{2\widehat{\mathbf{u}}_K^{\mathrm{T}}\widetilde{B}\widehat{\mathbf{u}}_{-K}}{\widehat{\mathbf{u}}_K^{\mathrm{T}}\widetilde{B}\widehat{\mathbf{u}}_K}\widehat{\epsilon}_{AB}^{\mathrm{T}}\widehat{\mathbf{u}}_K \\
&= \widehat{\mathbf{x}}^{\mathrm{T}}\widetilde{A}\widehat{\mathbf{x}} - \widehat{\epsilon}^{\mathrm{T}}\widehat{\mathbf{u}}_{-K}.
\end{aligned}$$

Here $\widehat{\epsilon} := \widetilde{A}\widehat{\mathbf{u}}_{-K} + \widehat{\epsilon}_{AB} - 2\frac{\epsilon_{AB}^{\mathrm{T}}\widehat{\mathbf{u}}_K}{\widehat{\mathbf{u}}_K^{\mathrm{T}}\widehat{B}\widehat{\mathbf{u}}_K}\widetilde{B}\widehat{\mathbf{u}}_K$.

For $\widehat{\epsilon}_{AB}$, we know that

$$\begin{aligned}
\|\widehat{\epsilon}_{AB}\|_2 &= \|\widetilde{A}\widehat{\mathbf{u}}_K - \widetilde{\lambda}_1\widetilde{B}\widehat{\mathbf{u}}_K\|_2 \\
&= \|\widetilde{A}\widehat{\mathbf{u}}_K - \widetilde{A}\widetilde{\mathbf{u}}_1 + \widetilde{\lambda}_1\widetilde{B}\widetilde{\mathbf{u}}_1 - \widetilde{\lambda}_1\widetilde{B}\widehat{\mathbf{u}}_K\|_2 \\
&= \|\widetilde{A}\widehat{\mathbf{u}}_K - \widetilde{A}\widetilde{\mathbf{u}}_1\|_2 + \|\widetilde{\lambda}_1\widetilde{B}\widetilde{\mathbf{u}}_1 - \widetilde{\lambda}_1\widetilde{B}\widehat{\mathbf{u}}_K\|_2 \\
&\leq (\|\widetilde{A}\|_2 + \widetilde{\lambda}\|\widetilde{B}\|_2)\|\widehat{\mathbf{u}}_K - \widetilde{\mathbf{u}}_1\|_2.
\end{aligned}$$

We can further bound $\|\widehat{\mathbf{u}}_K - \widetilde{\mathbf{u}}_1\|_2$ by $\|\widehat{\mathbf{u}}_K - \widehat{\mathbf{x}}\|_2 + \|\widehat{\mathbf{x}} - \mathbf{u}_{1r}\|_2 + \|\mathbf{u}_{1r} - \widetilde{\mathbf{u}}_1\|_2$ which is $O(\epsilon)$. Thus,

$$\begin{aligned}
\|\widehat{\epsilon}\|_2 &= \|\widetilde{A}\widehat{\mathbf{u}}_{-K} + \widehat{\epsilon}_{AB} - 2\frac{\widehat{\epsilon}_{AB}^{\mathrm{T}}\widehat{\mathbf{u}}_K}{\widehat{\mathbf{u}}_K^{\mathrm{T}}\widehat{B}\widehat{\mathbf{u}}_K}\widetilde{B}\widehat{\mathbf{u}}_K\|_2 \\
&\leq \|\widetilde{A}\|_2\|\widehat{\mathbf{u}}_{-K}\|_2 + \|\widehat{\epsilon}_{AB}\|_2 + 2\|\widetilde{B}^{1/2}\|_2\|\widehat{\mathbf{u}}_K\|_2\|\widehat{\epsilon}_{AB}\|_2 \\
&\leq \|\widetilde{A}\|_2\|\widehat{\mathbf{u}}_{-K}\|_2 + (1 + 2\|\widetilde{B}\|_2\|\widetilde{B}^{-1}\|_2)\|\widehat{\epsilon}_{AB}\|_2, \tag{62}
\end{aligned}$$

which is $O(\epsilon)$. Therefore, vector $\widehat{\mathbf{u}}_0$ satisfies both (60) and (61).

Observe that

$$-\widehat{\mathbf{x}}^{\mathrm{T}}\widetilde{A}\widehat{\mathbf{x}} + p_\lambda(\widehat{\mathbf{x}})$$
$$\geq \quad -\widehat{\mathbf{u}}_0^{\mathrm{T}}\widetilde{A}\widehat{\mathbf{u}}_0 - C\sqrt{\epsilon}\|\widehat{\mathbf{u}}_{-K}\|_2 + p_\lambda(\widehat{\mathbf{x}})$$
$$= \quad -\widehat{\mathbf{u}}_0^{\mathrm{T}}\widetilde{A}\widehat{\mathbf{u}}_0 - C\sqrt{\epsilon}\|\widehat{\mathbf{u}}_{-K}\|_2 + p_\lambda(\widehat{\mathbf{u}}_0) + p_\lambda(\widehat{\mathbf{u}}_{-K}) \tag{63}$$
$$\geq \quad -\widehat{\mathbf{u}}_0^{\mathrm{T}}\widetilde{A}\widehat{\mathbf{u}}_0 - C\sqrt{\epsilon}\|\widehat{\mathbf{u}}_{-K}\|_2 + p_\lambda(\widehat{\mathbf{u}}_0) + c\lambda\|\widehat{\mathbf{u}}_{-K}\|_1 \tag{64}$$
$$\geq \quad -\widehat{\mathbf{u}}_0^{\mathrm{T}}\widetilde{A}\widehat{\mathbf{u}}_0 + p_\lambda(\widehat{\mathbf{u}}_0), \tag{65}$$

where (63) uses the fact that $p_\lambda(\widehat{\mathbf{u}}_0[\mathcal{S}]) = p_\lambda(\widehat{\mathbf{x}}[\mathcal{S}])$; (64) uses the local equivalence between $p_\lambda$ and L1 norm; (65) holds when $\lambda \gg \sqrt{\epsilon}$. The strict equality holds as long as $\widehat{\mathbf{u}}_{-K} \equiv \mathbf{0}$. In other words, $\widehat{\mathbf{x}}$ should have the same support as $\mathbf{u}_1$. This completes the proof.

**Proof of Theorem 5** By Theorem 4, we know that the support of $\hat{x}$ is exactly equal to $\mathcal{S}$. In addition, by Theorem 3, we know that the estimator is consistent. It implies that $p_\lambda(\hat{x}) \equiv p_\lambda(\mathbf{u}_1)$. Therefore, $\hat{x}$ is equal to $\arg\min_{\mathbf{x}:\mathbf{x}[-\mathcal{S}]=\mathbf{0}} \mathbf{x}^{\mathrm{T}}\tilde{A}\mathbf{x}$ subject to $\mathbf{x}^{\mathrm{T}}\tilde{B}\mathbf{x} \leq 1$. In other words, $\hat{x}[\mathcal{S}]$ is the leading eigenvector of submatrix pair $(\tilde{A}_\mathcal{S}, \tilde{B}_\mathcal{S})$. By applying Theorem 1, we conclude the proof.

## 12 Proof of Results in Section 7

The following lemma from [35] characterizes the curvature of objective function.

**Lemma 3** *Suppose matrix $A$ has the following spectral decomposition $U_1\Lambda_1 U_1^{\mathrm{T}} + U_2\Lambda_2 U_2^{\mathrm{T}}$ with $U_1 \in \mathcal{O}(p, r)$ and $U_2 \in \mathcal{O}(p, p - r)$. (Here $\mathcal{O}(p, d)$ is the space of $p$ by $d$ orthonormal matrices.) Let $\delta := \min_j \Lambda_1[j, j] - \max_j \Lambda_2[j, j]$. Then it holds that*

$$\mathrm{tr}(A^{\mathrm{T}}(U_1 U_1^{\mathrm{T}} - X)) \geq \frac{1}{2}\delta\|U_1 U_1^{\mathrm{T}} - X\|_F^2, \tag{66}$$

*where $X$ is any matrix satisfying $\|X\|_2 \leq 1$ and $\|X\|_* \leq r$.*

**Proof of Theorem 7** We first define a few more auxiliary quantities, $\bar{U}_1 = U_1(U_1^{\mathrm{T}}\widetilde{B}U_1)^{-1/2}$, $\bar{X} = \bar{U}_1\bar{U}_1^{\mathrm{T}}$. and $\bar{\Lambda}_1 = (U_1^{\mathrm{T}}\widetilde{B}U_1)^{1/2}\Lambda_1(U_1^{\mathrm{T}}\widetilde{B}U_1)^{1/2}$. Next we give the upper bound for $\|\bar{U}_1 - U_1\|_2$, $\|\bar{X} - X\|_2$ and $\|\bar{\Lambda}_1 - \Lambda_1\|_2$. Specifically, we have

$$\|\bar{U}_1 - U_1\|_2 = \|U_1(U_1^{\mathrm{T}}\widetilde{B}U_1)^{-1/2} - U_1\|_2$$
$$\leq \|U_1\|_2\|(U_1^{\mathrm{T}}\widetilde{B}U_1)^{1/2} - I\|_2\|(U_1^{\mathrm{T}}\widetilde{B}U_1)^{-1/2}\|_2$$
$$\leq C\|U_1\|_2\|U_1^{\mathrm{T}}\widetilde{B}U_1 - I\|_2\|(U_1^{\mathrm{T}}\widetilde{B}U_1)^{-1/2}\|_2,$$

$$\|\bar{X} - X\|_2 = \|\bar{U}_1\bar{U}_1^{\mathrm{T}} - U_1 U_1^{\mathrm{T}}\|$$
$$= \|\bar{U}_1\|_2\|\bar{U}_1 - U_1\|_2 + \|U_1\|_2\|\bar{U}_1 - U_1\|_2,$$

and

$$\|\bar{\Lambda}_1 - \Lambda_1\|_2 = \|(U_1^{\mathrm{T}}\widetilde{B}U_1)^{1/2}\Lambda_1(U_1^{\mathrm{T}}\widetilde{B}U_1)^{1/2} - \Lambda_1\|_2$$
$$\leq \|(U_1^{\mathrm{T}}\widetilde{B}U_1)^{1/2} - I\|_2\|\Lambda_1(U_1^{\mathrm{T}}\widetilde{B}U_1)^{1/2}\|_2 + \|\Lambda_1\|_2\|(U_1^{\mathrm{T}}\widetilde{B}U_1)^{1/2} - I\|_2$$
$$\leq C((U_1^{\mathrm{T}}\widetilde{B}U_1)^{1/2}\|_2 + 1)\|\Lambda_1\|_2\|(U_1^{\mathrm{T}}\widetilde{B}U_1) - I\|_2.$$

We construct $\bar{A} = \widetilde{B}U_1\Lambda_1 U_1^{\mathrm{T}}\widetilde{B}$. Then the infinity norm $\|\bar{A} - \widetilde{A}\|_\infty$ can be bounded by

$$\|\bar{A} - \widetilde{A}\|_\infty \leq \|\bar{A} - A\|_\infty + \|\widetilde{A} - A\|_\infty$$
$$= \|\widetilde{B}U_1\Lambda_1 U_1^{\mathrm{T}}\widetilde{B} - BU_1\Lambda_1 U_1^{\mathrm{T}}B\|_\infty + \|\widetilde{A} - A\|_\infty$$
$$\leq \|(\widetilde{B} - B)U_1\Lambda_1 U_1^{\mathrm{T}}\widetilde{B}\|_\infty + \|BU_1\Lambda_1 U_1^{\mathrm{T}}(\widetilde{B} - B)\|_\infty + \|\widetilde{A} - A\|_\infty$$
$$\leq \lambda_1\|B^{-1/2}\|\|B^{1/2}\|(\|B\|_\infty + \|\widetilde{B}\|_\infty)\sqrt{|\mathcal{S}|}\|\widetilde{B} - B\|_\infty + \|\widetilde{A} - A\|_\infty, \tag{67}$$

where the last inequality uses $\|BU_1\Lambda_1 U_1^T(\widetilde{B}-B)\|_\infty \le \lambda_1\|B^{-1/2}\|\|B^{1/2}\|\|B\|_\infty\sqrt{|\mathcal{S}|}\|\widetilde{B}-B\|_\infty$ depending on the fact that $\|U_1\Lambda_1 U_1^T B[:,j]\|_2$ are bounded for each $j$. This is because that $\|U_1^T B\|_2 \le \|B^{1/2}\|$. Thus $\|(U_1^T B)\mathbf{e}_j\|_2 \le \|B^{1/2}\|\|\mathbf{e}_j\|_2$, where $\mathbf{e}_j$ is an one-hot vector with $j$th entry being 1. Additionally, it is easy to see that $\Lambda_1$ is bounded by $\lambda_1$ and $\|U_1\|_1$ is bounded by $\|B^{-1/2}\|_2$. Thus $\|U_1\Lambda_1 U_1^T B[:,j]\|_2$ is bounded for each $j$.

By the optimality of $\widehat{X}$, we have

$$\mathrm{tr}(\widetilde{A}\widehat{X}) - p_\lambda(\widehat{X}) \ge \mathrm{tr}(\widetilde{A}\bar{X}) - p_\lambda(\bar{X}).$$

Let $\Delta = \widehat{X} - \bar{X}$ and make the rearrangement. The above equation can be written as

$$-\mathrm{tr}(\bar{A},\Delta) \le p_\lambda(\bar{X}) - p_\lambda(\bar{X}+\Delta) + \mathrm{tr}((\widetilde{A}-\bar{A})\Delta). \tag{68}$$

Define the support set $\mathcal{T} = \mathcal{S}\times\mathcal{S}\in [p]\times[p]$. For the first term of (68),

$$
\begin{aligned}
p_\lambda(\bar{X}) - p_\lambda(\bar{X}+\Delta) &= p_\lambda(\bar{X}[\mathcal{T}]) - p_\lambda(\bar{X}[\mathcal{T}]+\Delta[\mathcal{T}]) - p_\lambda(\Delta[\mathcal{T}^c]) \\
&\le p_\lambda(\Delta[\mathcal{T}]) - p_\lambda(\Delta[\mathcal{T}]^c).
\end{aligned}
$$

For the second term of (68), we have $\mathrm{tr}((\widetilde{A}-\bar{A})\Delta) \le \|\widetilde{A}-\bar{A}\|_\infty\|\Delta\|_1$. Putting together, we have

$$-\mathrm{tr}(\bar{A},\Delta) \le p_\lambda(\Delta[\mathcal{T}]) - p_\lambda(\Delta[\mathcal{T}^c]) + \|\widetilde{A}-\bar{A}\|_\infty\|\Delta\|_1. \tag{69}$$

By the local equivalence between $p_\lambda$ and $\lambda|x|$, we have

$$-\mathrm{tr}(\bar{A},\Delta) \le c_2\lambda\|\Delta[\mathcal{T}_0]\|_1 - c_1\lambda\|\Delta[\mathcal{T}_0^c]\|_1 + \|\widetilde{A}-\bar{A}\|_\infty\|\Delta\|_1, \tag{70}$$

where $\mathcal{T}_0 = \mathcal{T}\cup\mathcal{T}_{large}$ with $\mathcal{T}_{large} = \{(i,j): |\Delta[i,j]| > \gamma\lambda\}$. We can observe that

$$|\mathcal{T}_{large}| \le C|\mathcal{S}|^2, \tag{71}$$

for some constant $C$. This fact will be proved later.

Next, by Lemma 3, observe that

$$
\begin{aligned}
-\mathrm{tr}(\widetilde{A},\Delta) &= \mathrm{tr}(\widetilde{A},\bar{X}-\widehat{X}) \\
&= \mathrm{tr}(\widetilde{B}U_1\Lambda_1 U_1^T\widetilde{B}, \bar{U}_1\bar{U}_1^T - \widehat{X}) \\
&= \mathrm{tr}(\widetilde{B}^{1/2}U_1\Lambda_1 U_1^T\widetilde{B}^{1/2}, \widetilde{B}^{1/2}\bar{U}_1\bar{U}_1^T\widetilde{B}^{1/2} - \widetilde{B}^{1/2}\widehat{X}\widetilde{B}^{1/2}) \\
&\ge \frac{\bar{\lambda}_r}{2}\|\widetilde{B}^{1/2}\Delta\widetilde{B}^{1/2}\|_F^2 \\
&\ge \frac{\lambda_r}{4}\|\widetilde{B}^{1/2}\Delta\widetilde{B}^{1/2}\|_F^2. \tag{72}
\end{aligned}
$$

The last inequality holds since $\bar{\lambda}$ and $\lambda$ are close. The right hand side of (70) is bounded by $2c_2\lambda\|\Delta[\mathcal{T}_0]\|_1$ which is further bounded by $2c_2|\mathcal{S}|\lambda\|\Delta[\mathcal{T}_0]\|_F$. By these facts, we have that

$$\lambda_r\|\widetilde{B}^{1/2}\Delta\widetilde{B}^{1/2}\|_F^2 \le 8c_2|\mathcal{S}|\lambda\|\Delta[\mathcal{T}_0]\|_F. \tag{73}$$

Additionally, (70) further gives the cone constraint.

$$\|\Delta[\mathcal{T}_0^c]\|_1 \le 4c_2/c_1\|\Delta[\mathcal{T}_0]\|_1. \tag{74}$$

Following the proof technique in the compress sensing [4], define the index set $\mathcal{T}_1 \subset ([p]\times[p]\backslash\mathcal{T}_0)$ which correspond to the entries with the largest absolute values in $\Delta$. Similarly, we define $\mathcal{T}_k$ $(k\ge 2)$ sequentially such that $\mathcal{T}_k$ is the set of indices corresponding to $t$ largest absolute values in $\Delta$ outside $\bigcup_{l<k}\mathcal{T}_l$. By triangle inequality, we have

$$
\begin{aligned}
\|\widetilde{B}^{1/2}\Delta\widetilde{B}^{1/2}\|_F &\ge \|\widetilde{B}^{1/2}\Delta[\mathcal{T}_{01}]\widetilde{B}^{1/2}\|_F - \sum_{k\ge 2}\|\widetilde{B}^{1/2}\Delta[\mathcal{T}_k]\widetilde{B}^{1/2}\|_F \\
&\ge \phi_{min}^{\widetilde{B}}(2s_u+t)\|\Delta[\mathcal{T}_{01}]\|_F - \phi_{max}^{\widetilde{B}}(t)\sum_{k\ge 2}\|\Delta[\mathcal{T}_k]\|_F.
\end{aligned}
$$

In above, we use the following quantities

$$\phi_{max}^B(k) := \max_{\|u\|_0 \le k, u \ne 0} \frac{u^{\mathrm{T}} B u}{u^{\mathrm{T}} u}, \quad \phi_{min}^B(k) := \min_{\|u\|_0 \le k, u \ne 0} \frac{u^{\mathrm{T}} B u}{u^{\mathrm{T}} u}, \tag{75}$$

which are known as the restricted eigenvalues for positive definite matrix $B$.

By the definition of $\mathcal{T}_k$ and cone constraint, we have

$$
\begin{aligned}
\sum_{k \ge 2} \|\Delta[\mathcal{T}_k]\|_F &\le \sqrt{t} \sum_{k \ge 2} \|\Delta[\mathcal{T}_k]\|_\infty \\
&\le \frac{1}{\sqrt{t}} \sum_{k \ge 1} \|\Delta[\mathcal{T}_1]\|_1 \\
&\le (4c_2)/(\sqrt{t}c_1)\|\Delta[\mathcal{T}_0]\|_1 \\
&\le (4Cc_2)/(\sqrt{t}c_1)|\mathcal{S}|\|\Delta[\mathcal{T}_{01}]\|_F.
\end{aligned}
\tag{76}
$$

Hence, $\|\widetilde{B}^{1/2}\Delta\widetilde{B}^{1/2}\|_F$ is lower bounded by $\kappa\|\Delta[\mathcal{T}_{01}]\|_F$ with

$$\kappa = \phi_{min}^{\widetilde{B}}(C|\mathcal{S}| + t) - (4Cc_2|\mathcal{S}|)/(\sqrt{t}c_1)\phi_{max}^{\widetilde{B}}(t). \tag{77}$$

Here $\kappa$ is a strictly positive constant when perturbation error $\epsilon$ is small enough. Together with (70), we have

$$\|\Delta[\mathcal{T}_{01}]\|_F \le C\frac{|\mathcal{S}|\lambda}{\kappa^2\lambda_r}. \tag{78}$$

Together with (76), we have

$$\|\Delta\|_F \le C\|\Delta[\mathcal{T}_{01}]\|_F \le C\frac{|\mathcal{S}|\lambda}{\kappa^2\lambda_r} \tag{79}$$

by choosing large $t$ and adjusting the constant $C$.

Now we go back to show the fact that $|\mathcal{T}_0|$ is bounded by $C|\mathcal{S}|^2$. We consider the following two situations.

1. When $\tilde{B}$ is not singular, then $\|\tilde{B}^{-1}\|$ are bounded by some constant $c_b$. Therefore, by (69) and (72), we have

$$\frac{\lambda_r}{4c_b}\|\Delta\|_F^2 \le \frac{1}{2}|\mathcal{S}|^2\gamma\lambda^2 - p_\lambda(\Delta[\mathcal{T}^c]) + \|\widetilde{A} - \bar{A}\|_\infty\|\Delta\|_1$$

Since $p_\lambda(x) > \|\widetilde{A} - \bar{A}\|_\infty |x|$ when $|x| \ge \gamma\lambda$, then we have

$$\frac{\lambda_r}{4c_b}\|\Delta[\mathcal{T}_0]\|_F^2 \le \frac{1}{2}|\mathcal{S}|^2\gamma\lambda^2 + \|\widetilde{A} - \bar{A}\|_\infty\|\Delta[\mathcal{T}_0]\|_1.$$

Notice that each element of $|\Delta[\mathcal{T}_0]|$ is at least $\gamma\lambda$. Then $\frac{\lambda_r}{4c_b}\|\Delta[\mathcal{T}_0]\|_F^2 - \|\widetilde{A} - \bar{A}\|_\infty\|\Delta[\mathcal{T}_0]\|_1$ is lower bounded by $|\mathcal{T}_0| \cdot \gamma^2\lambda^2$ times some constant. Then we must have $|\mathcal{T}_0| \cdot \gamma^2\lambda^2 \le \frac{1}{2}|\mathcal{S}|^2\gamma\lambda^2$. It implies that $|\mathcal{T}_{large}| \le C|\mathcal{S}|^2$. So is $|\mathcal{T}_0|$.

2. When $\tilde{B}$ is singular, by using condition that $e_{approx} \le \lambda$, we have

$$p_\lambda(\hat{X}) \le e_{approx}^2 + \frac{1}{2}C|\mathcal{S}|^2\lambda^2. \tag{80}$$

Thus we know there are at most $C|\mathcal{S}|^2$ entries in $|\hat{X}|$ are larger than $\gamma\lambda$. It indicates $|\mathcal{T}_{large}| \le C|\mathcal{S}|^2$. Therefore, we have proved (71).

**Byproduct: space perturbation** We know $\sigma_{r+1}(\bar{X}) = 0$ and

$$\sigma(\widehat{X}) \ge \sigma(UU^T) - \|\widehat{X} - \bar{X}\|_2 - \|\bar{U}\bar{U}^T - UU^T\|_2.$$

Thus $\sigma(\widehat{X})$ is lower bounded by some positive constant. By Wedin $\sin\theta$ theorem, we then have

$$\|P_{\widehat{U}_1} - P_{U_1}\|_F = \|P_{\widehat{U}_1} - P_{\bar{U}_1}\|_F \le \frac{2\|\widehat{X} - \bar{X}\|_F}{\sigma_r(\widehat{X}) - \sigma_{r+1}(\bar{X})} \le C\|\widehat{X} - \bar{X}\|_F, \tag{81}$$

by adjusting the constant $C$. Here $P_X$ is the projection matrix on the column space spanned by $X$.

# 13 Proofs of Propositions 1 and 2

We provide the proofs of Propositions 1 and 2 in this section.

**Proposition 1** *Let $\check{\mathbf{y}}$ be the projection of $\mathbf{y}$ on to the ellipsoid $\{\mathbf{x} \mid \mathbf{x}^T B \mathbf{x} = 1\}$. Then $\check{\mathbf{y}}$ has the following form*

$$\check{\mathbf{y}} = (\beta B + I)^{-1} \mathbf{y},$$

*where $\beta$ is a scalar which is the solution to the equation $1 = \sum_j \frac{d_j (\widetilde{\mathbf{y}}[j])^2}{(\beta d_j + 1)^2}$, where $\widetilde{\mathbf{y}} = U^T \mathbf{y}$; $B = U D U^T$ and $D = \text{diag}(d_1, \ldots, d_p)$.*

**Proof of Proposition 1** Since $\check{\mathbf{y}}$ is the projection of $\mathbf{y}$ is onto the ellipsoid, then we must have that $\check{\mathbf{y}} - \mathbf{y}$ is orthogonal to the tangent space of ellipsoid at $\check{\mathbf{y}}$. By straightforward calculation, we know the normal direction of the tangent space at $\check{\mathbf{y}}$ is $B\check{\mathbf{y}}$. Thus, we must have that

$$\mathbf{y} - \check{\mathbf{y}} \propto B\check{\mathbf{y}}.$$

In other words,

$$\mathbf{y} = (\beta B + I)\check{\mathbf{y}}$$

holds for some constant $\beta$. Notice that $\check{\mathbf{y}}$ is on the surface of the ellipsoid. Then

$$
\begin{aligned}
1 &= \mathbf{y}^T (\beta B + I)^{-1} B (\beta B + I)^{-1} \mathbf{y} \\
&= \mathbf{y}^T U (\beta D + I)^{-1} D (\beta D + I)^{-1} U^T \mathbf{y},
\end{aligned}
$$

where $B$ has eigen-decomposition $B = U D U^T$. Therefore, we have

$$1 = \sum_j \frac{d_j (\widetilde{\mathbf{y}}[j])^2}{(\beta d_j + 1)^2},$$

with $\widetilde{\mathbf{y}} = U^T \mathbf{y}$. This concludes the proof.

**Proposition 2** *The limiting point returned by (16) is the stationary point of sub-problem (15).*

**Proof of Proposition 2** Notice that the objective

$$\arg\min_{\mathbf{x} \in \mathcal{D}} \quad \frac{\eta}{2} \|\mathbf{x} - \mathbf{z}^{(t)}\|_2^2 + (\mathbf{y}^{(t)})^T (\mathbf{x} - \mathbf{z}^{(t)}) - \mathbf{x}^T \widetilde{A} \mathbf{x} \tag{82}$$

can be rewritten as

$$\arg\min_{\mathbf{x} \in \mathcal{D}, \mathbf{v} \in \mathcal{V}} \quad \frac{\eta}{2} \|\mathbf{x} - \mathbf{z}^{(t)}\|_2^2 + (\mathbf{y}^{(t)})^T (\mathbf{x} - \mathbf{z}^{(t)}) - \mathbf{v}^T X_h \mathbf{x} \tag{83}$$

where $X_h^T X_h = \widetilde{A}$ and $\mathcal{V} = \{\mathbf{v} : \|\mathbf{v}\| \leq \|X_h \mathbf{x}\|\}$. If $\check{\mathbf{x}}$ is the solution to (82), then $(\check{\mathbf{x}}, \check{\mathbf{v}})$ is the solution to (83) where $\check{\mathbf{v}} = X_h \check{\mathbf{x}}$. Therefore, we only need to solve (83). We iteratively optimize with respect to $\mathbf{v}$ and $\mathbf{x}$ and get the following update rule,

$$
\begin{aligned}
b_m^{(t+1)} &= \mathbf{z}^{(t)} - \frac{\mathbf{y}^{(t)} - X_h^T v_{m-1}^{(t+1)}}{\eta}, \\
\mathbf{x}_m^{(t+1)}[\mathcal{S}_b] &= (\beta_m^{(t+1)} \widetilde{B}_{\mathcal{S}_b} + I)^{-1} b_m^{(t+1)}[\mathcal{S}_b], \\
\mathbf{v}_m^{(t+1)} &= X_h \mathbf{x}_m^{(t+1)}.
\end{aligned}
\tag{84}
$$

By simplification, we then obtain (16). This concludes the proof.

# 14 Convergence of NC-SGEP

Recall our optimization problem,

$$\min_{\mathbf{x}, \mathbf{y}, \mathbf{v}, \mathbf{z}} \quad -\mathbf{x}^T \tilde{A} \mathbf{x} + p_\lambda(\mathbf{z}) + \mathbf{y}^T(\mathbf{x} - \mathbf{z}) + \frac{\eta}{2} \|\mathbf{x} - \mathbf{z}\|_2^2$$

$$+ \infty \mathbf{1}\{\mathbf{x}^T \widetilde{B} \mathbf{x} > 1\} + \infty \mathbf{1}\{\|\mathbf{x}\|_0 > s_n\}, \tag{85}$$

we can write our Lagrangian function $\mathcal{L}(\mathbf{x}, \mathbf{z}, \mathbf{y})$ as

$$\mathcal{L}(\mathbf{x}, \mathbf{z}, \mathbf{y}) = g(\mathbf{x}) + h(\mathbf{z}) + \mathbf{y}^T(\mathbf{x} - \mathbf{z}) + \frac{\eta}{2} \|\mathbf{x} - \mathbf{z}\|^2, \tag{86}$$

where $g(\mathbf{x}) = -\mathbf{x}^T \tilde{A} \mathbf{x}$ is a $L$-Lipschitz smooth function over domain $\mathcal{D} = \{\mathbf{x} : \mathbf{x} : \|\mathbf{x}\|_0 \leq s_n; \mathbf{x}^T \widetilde{B} \mathbf{x} = 1\}$ for some constant $L$, and $h(\mathbf{z}) = p_\lambda(\mathbf{z})$ is a non-smooth function with a convex domain. Here constant $L$ can be bounded by norm $\|(\tilde{B}[\mathcal{S}, \mathcal{S}])^{-1/2} \tilde{A}[\mathcal{S}, \mathcal{S}](\tilde{B}[\mathcal{S}, \mathcal{S}])^{-1/2}\|_2$ for any subset $\mathcal{S}$ with $|\mathcal{S}| \leq s_n$.

Furthermore, we know that both $g(\mathbf{x}) + \frac{\kappa_0}{2}\|\mathbf{x}\|^2$ and $h(\mathbf{x}) + \frac{\kappa_0}{2}\|\mathbf{x}\|^2$ are strongly convex for a large constant $\kappa_0$. By the initial condition, we have that $\mathbf{y}^0 = 0$, $\mathbf{x}^0 = \mathbf{z}^0$, and $\mathrm{supp}(\mathbf{x}^0) = \mathcal{S}_1$ contains the true support set. By solving $\mathbf{x}^1$, we can see that $\mathrm{supp}(\mathbf{x}^1) = \mathrm{supp}(\mathbf{x}^0) = \mathrm{supp}(\mathbf{z}^{(0)})$. Then, $\mathbf{z}^1$ is again a sparse solution with $\mathrm{supp}(\mathbf{u}_1) \subset \mathrm{supp}(\mathbf{z}^1) \subset \mathrm{supp}(\mathbf{z}^0)$. By repeating this procedure, we know that $\mathrm{supp}(\mathcal{S}_t) = \mathrm{supp}(\mathcal{S}_{t-1}) = \mathrm{supp}(\mathcal{S}_1)$. Thus in the following, we only need to work on the restricted space $\{\mathbf{x} : \mathbf{x}^T \widetilde{B} \mathbf{x} = 1, \mathbf{x}[-\mathcal{S}_1] = \mathbf{0}\}$. Then it becomes fixed-support ADMM algorithm without worrying about the changing support issue.

We then can show that

$$\|\mathbf{y}^{t+1} - \mathbf{y}^t\| \leq L\|\mathbf{x}^{t+1} - \mathbf{x}^t\|. \tag{87}$$

This is because, by the optimality of $\mathbf{x}^{t+1}$, we have

$$\nabla g(\mathbf{x}^{t+1}) + \mathbf{y}^t + \eta(\mathbf{x}^{t+1} - \mathbf{z}^{t+1}) = \mathbf{0}.$$

By noticing the update formula that $\mathbf{y}^{t+1} = \mathbf{y}^{(t)} + \eta(\mathbf{x}^{t+1} - \mathbf{z}^{t+1})$, we get

$$\nabla g(\mathbf{x}^{t+1}) = -\mathbf{y}^{t+1}.$$

Therefore,

$$\|\mathbf{y}^{t+1} - \mathbf{y}^t\| = \|\nabla g(\mathbf{x}^{t+1}) - \nabla g(\mathbf{x}^t)\| \leq L\|\mathbf{x}^{t+1} - \mathbf{x}^t\|,$$

and it leads to (87).

Next, we show that

$$\mathcal{L}(\mathbf{x}^{t+1}, \mathbf{z}^{t+1}, \mathbf{y}^{t+1}) - \mathcal{L}(\mathbf{x}^t, \mathbf{z}^t, \mathbf{y}^t) \leq \left(\frac{L^2}{\eta} - \frac{\gamma}{2}\right)\|\mathbf{x}^{t+1} - \mathbf{x}^t\|^2 - \frac{\gamma}{2}\|\mathbf{z}^{t+1} - \mathbf{z}^t\|^2, \tag{88}$$

where $\gamma$ is a positive constant which will be explained later.

We split the successive difference of Lagrangian function by

$$\mathcal{L}(\mathbf{x}^{t+1}, \mathbf{z}^{t+1}, \mathbf{y}^{t+1}) - \mathcal{L}(\mathbf{x}^t, \mathbf{z}^t, \mathbf{y}^t)$$
$$= \mathcal{L}(\mathbf{x}^{t+1}, \mathbf{z}^{t+1}, \mathbf{y}^{t+1}) - \mathcal{L}(\mathbf{x}^{t+1}, \mathbf{z}^{t+1}, \mathbf{y}^t) + \mathcal{L}(\mathbf{x}^{t+1}, \mathbf{z}^{t+1}, \mathbf{y}^t) - \mathcal{L}(\mathbf{x}^t, \mathbf{z}^t, \mathbf{y}^t). \tag{89}$$

The first term of (89) can be bounded by

$$\mathcal{L}(\mathbf{x}^{t+1}, \mathbf{z}^{t+1}, \mathbf{y}^{t+1}) - \mathcal{L}(\mathbf{x}^{t+1}, \mathbf{z}^{t+1}, \mathbf{y}^t)$$
$$= \langle \mathbf{y}^{t+1} - \mathbf{y}^t, \mathbf{x}^{t+1} - \mathbf{z}^{t+1} \rangle$$
$$= \frac{1}{\eta}\|\mathbf{y}^{t+1} - \mathbf{y}^t\|^2$$
$$\leq \frac{L^2}{\eta}\|\mathbf{x}^{t+1} - \mathbf{x}^t\|^2. \tag{90}$$

The second term of (89) can be bounded by

$$\mathcal{L}(\mathbf{x}^{t+1}, \mathbf{z}^{t+1}, \mathbf{y}^t) - \mathcal{L}(\mathbf{x}^t, \mathbf{z}^t, \mathbf{y}^t)$$
$$= \mathcal{L}(\mathbf{x}^{t+1}, \mathbf{z}^{t+1}, \mathbf{y}^t) - \mathcal{L}(\mathbf{x}^t, \mathbf{z}^{t+1}, \mathbf{y}^t) + \mathcal{L}(\mathbf{x}^t, \mathbf{z}^{t+1}, \mathbf{y}^t) - \mathcal{L}(\mathbf{x}^t, \mathbf{z}^t, \mathbf{y}^t)$$
$$\leq \langle \nabla \mathcal{L}(\mathbf{x}^{t+1}, \mathbf{z}^{t+1}, \mathbf{y}^t), \mathbf{x}^{t+1} - \mathbf{x}^t \rangle - \frac{\gamma}{2} \|\mathbf{x}^{t+1} - \mathbf{x}^t\|^2$$
$$+ \langle \xi^{t+1}, \mathbf{z}^{t+1} - \mathbf{z}^t \rangle - \frac{\gamma}{2} \|\mathbf{z}^{t+1} - \mathbf{z}^t\|^2 \tag{91}$$
$$\leq -\frac{\gamma}{2} \|\mathbf{x}^{t+1} - \mathbf{x}^t\|^2 - \frac{\gamma}{2} \|\mathbf{z}^{t+1} - \mathbf{z}^t\|^2, \tag{92}$$

where (91) holds since the $\mathcal{L}$ is a $\gamma$-strongly convex function of $\mathbf{x}_1$ and $\mathbf{x}_0$ by assumption for some constant $\gamma$. (This is true since we can take $\eta$ large enough to make the lagrangian function strictly convex. $\gamma$ may depend on $\eta$ and $\gamma \geq \eta - L$.) and (92) holds since that $\langle \nabla \mathcal{L}(\mathbf{x}^{t+1}, \mathbf{z}^{t+1}, \mathbf{y}^t), \mathbf{x}^{t+1} - \mathbf{x}^t \rangle = 0$ and $\langle \xi^{t+1}, \mathbf{z}^{t+1} - \mathbf{z}^t \rangle \leq 0$ by the optimality of $\mathbf{x}_1^{t+1}$, $\mathbf{x}_0^t$ and convexity of the domain. Here $\xi^{t+1}$ is the subdifferential of $\mathcal{L}(\mathbf{x}^t, \mathbf{z}, \mathbf{y}^t)$ at $\mathbf{z}^{t+1}$. Summing (90) and (92), we get the desired result, i.e., (88) holds.

Third, we show that $\mathcal{L}(\mathbf{x}^t, \mathbf{z}^t, \mathbf{y}^t)$ is lower bounded by some constant $\underline{f}$. By definition of $\mathcal{L}(\mathbf{x}^{t+1}, \mathbf{z}^{t+1}, \mathbf{y}^{t+1})$, we have

$$\mathcal{L}(\mathbf{x}^{t+1}, \mathbf{z}^{t+1}, \mathbf{y}^{t+1})$$
$$= h(\mathbf{z}^{t+1}) + (g(\mathbf{x}^{t+1}) + \langle \mathbf{y}^{t+1}, \mathbf{x}^{t+1} - \mathbf{z}^{t+1} \rangle) + \frac{\eta}{2} \|\mathbf{x}^{t+1} - \mathbf{z}^{t+1}\|^2$$
$$= h(\mathbf{z}^{t+1}) + (g(\mathbf{x}^{t+1}) + \langle \nabla g(\mathbf{x}^{t+1}), \mathbf{z}^{t+1} - \mathbf{x}^{t+1} \rangle) + \frac{\eta}{2} \|\mathbf{x}^{t+1} - \mathbf{z}^{t+1}\|^2 \tag{93}$$
$$\geq h(\mathbf{z}^{t+1}) + g(\mathbf{z}^{t+1}) \tag{94}$$
$$= f(\mathbf{z}^{t+1}) \geq \underline{f}. \tag{95}$$

Here, (93) uses the fact that $\mathbf{y}^{t+1} = -\nabla g(\mathbf{x}^{t+1})$ and (94) uses the Lipschitz continuity of $g$ when $\eta \geq L$. Lastly, (95) holds since $\underline{f} := \min_{\mathbf{x}} h(\mathbf{x}) + g(\mathbf{x})$ is the lower bound of the objective function.

Therefore, by above results, we know that $\mathcal{L}(\mathbf{x}^t, \mathbf{z}^t, \mathbf{y}^t)$ monotonically decreases and converges to some limit. In fact, we can further show that

$$\lim_{t \to 0} \|\mathbf{x}^t - \mathbf{z}^t\| \to 0. \tag{96}$$

Since we know that

$$\sum_{t=1}^{T} \left\{ (\frac{\gamma}{2} - \frac{L^2}{\eta}) \|\mathbf{x}^{t+1} - \mathbf{x}^t\|^2 + \frac{\gamma}{2} \|\mathbf{z}^{t+1} - \mathbf{z}^t\|^2 \right\} < \infty \tag{97}$$

and both $\frac{\gamma}{2} - \frac{L^2}{\eta}$ and $\gamma$ are positive, then it immediately leads to that

$$\lim_t \|\mathbf{x}^{t+1} - \mathbf{x}^t\| \to 0 \text{ and } \lim_t \|\mathbf{z}^{t+1} - \mathbf{z}^t\| \to 0.$$

Since $L\|\mathbf{x}^{t+1} - \mathbf{x}^t\| \geq \|\mathbf{y}^{t+1} - \mathbf{y}^t\|$, it further gives

$$\lim_t \|\mathbf{y}^{t+1} - \mathbf{y}^t\| \to 0.$$

In addition, note that $\mathbf{y}^{t+1} - \mathbf{y}^t = \eta(\mathbf{x}^{t+1} - \mathbf{z}^{t+1})$, thus

$$\lim_t \|\mathbf{x}^t - \mathbf{z}^t\| \to 0,$$

this concludes the proof of (96).

Recall the definition that $T(\epsilon)$ be the minimum $t$ such that $\|\mathbf{x}^t - \mathbf{z}^t\| \leq \epsilon$. Therefore, for any $t \in \{1, \ldots, T(\epsilon)\}$, we have $\|\mathbf{x}^t - \mathbf{z}^t\| > \epsilon$. Hence, we know $\|\mathbf{x}^{t+1} - \mathbf{x}^t\| > \eta\epsilon/L$. Thus by (88), it leads to

$$\mathcal{L}(\mathbf{x}^{t+1}, \mathbf{z}^{t+1}, \mathbf{y}^{t+1}) - \mathcal{L}(\mathbf{x}^t, \mathbf{z}^t, \mathbf{y}^t) \leq (\frac{L^2}{\eta} - \gamma/2) \frac{\eta\epsilon}{L}.$$

Therefore, we arrive at

$$T(\epsilon) \leq \frac{(\mathcal{L}(\mathbf{x}^0, \mathbf{z}^0, \mathbf{y}^0) - \underline{f})}{(\gamma/(2L) - L/\eta)\eta\epsilon}.$$

By taking $\gamma = \eta - L$, we will have

$$T(\epsilon) \leq \frac{(\mathcal{L}(\mathbf{x}^0, \mathbf{z}^0, \mathbf{y}^0) - \underline{f})}{(\eta/(2L) - L/\eta - \frac{1}{2})\eta\epsilon}.$$

This completes the proof.

## 15 Estimation Procedure for Semidefinite Programming

For computation, (P2) can be equivalently written as

$$\min_{X,Y} \quad -\mathrm{tr}(AX) + p_\lambda(X) + \infty\mathbf{1}\{\|Y\|_* > r\} + \infty\mathbf{1}\{\|Y\|_2 > 1\},$$
$$s.t. \quad B^{1/2}XB^{1/2} = Y. \tag{98}$$

Then the corresponding augmented Lagrangian multiplier problem is

$$\min_{X,Y} \quad -\mathrm{tr}(AX) + p_\lambda(X) + \infty\mathbf{1}\{\|Y\|_* > r\} + \infty\mathbf{1}\{\|Y\|_2 > 1\},$$
$$+\mathrm{tr}(Z^T(B^{1/2}XB^{1/2} - Y)) + \frac{\eta}{2}\|B^{1/2}XB^{1/2} - Y\|_F^2 \tag{99}$$

The procedure for estimation of $\hat{X}$ via SDP can be summarized as follows. Based on (99), we aim to solve $X, Y$ and $Z$ iteratively.

a. Update $Y$: Compute $K^{(t)} := X^{(t)} + (A + Z^{(t)})/\eta$ and symmetrize $K^{(t)} = \frac{1}{2}(K^{(t)} + (K^{(t)})^T)$. Do the projection $Y^{(t+1)} := \mathrm{Proj}_{\mathcal{C}_{fan}}(K^{(t)})$.

b. Update $X$: Compute $H^{(t)} := Y^{(t)} - Z^{(t)}/\eta$. Solve the regular penalized linear regression problem
$$\mathbf{x}^{(t+1)} := \arg\min_{\mathbf{x}} \|B_B\mathbf{x} - \mathbf{h}^{(t)}\|_2^2 + p_\lambda(\mathbf{x}),$$
where $\mathbf{x}, \mathbf{h}^{(t)} \in \mathbb{R}^{p^2}$ are the vectorized versions of $X$ and $H^{(t)}$, respectively. $B_B := B^{1/2} \otimes B^{1/2}$ is the outer product of two $B^{1/2}$s. Lastly, transfer $\mathbf{x}^{(t+1)}$ to the matrix form to get $X^{(t+1)}$.

c. Update $Z$: $Z^{(t+1)} = Z^{(t)} + \eta(X^{(t)} - Y^{(t)})$.

According to Lemma 4.1 in [35], for an arbitrary matrix $X$, we can find that

$$\mathrm{Proj}_{\mathcal{C}_{fan}}(X) = \sum_{j=1}^{p} \gamma_j(\theta)\mathbf{v}_j\mathbf{v}_j^T, \tag{100}$$

where $X$ admits eigen-decomposition $X = \sum_{j=1}^{p} \gamma_j\mathbf{v}_j\mathbf{v}_j^T$; $\gamma_j(\theta) = \min\{\max\{\gamma_j - \theta, 0\}, 1\}$ and $\theta$ is the solution to $\sum_j \gamma_j(\theta) = r$.