# OpenReview forum: "A Note on Sparse Generalized Eigenvalue Problem"
_NeurIPS.cc/2021/Conference — NeurIPS 2021 Poster_

### Official Review · Reviewer_bjrj · 2021-07-07

**Rating:** 6
**Confidence:** 4

**Summary:**

The paper is dedicated to the sparse generalised eigenvalue problem (SGEP), which consists in recovering the leading generalized eigenvector corresponding to the largest generalized eigenvalue of the matrix pair (A,B). The authors consider the noisy version of A and B: \tilde{A} and \tilde{B}. The main challenges are: problem NP hard because of the subset selection problem (the sparse support size is unknown); the noisy matrices may be ill-conditioned to that algorithms requiring their inverse may not work.
The contributions of the authors are as follows: derivation of an error bound between the true sparse eigenvector and the solution of the SGEP; derivation of the conditions for support recovery for potentially non-convex penalty functions; an ADMM based algorithm is proposed to solve the SGEP.

**Limitations And Societal Impact:**

I think the authors proposed a interesting extension of SGEP problem with non-convex regularisation and their method is supported by some theoretical results. Please see my comments above concerning the limitations.

**Main Review:**

The SGEP is an important tool for data analysis and is relevant when facing high-dimensional problems. I found the paper interesting regarding the issues raised by the authors and the properties. That being said, I would have several comments and hope they will be useful.
- On the clarity:
1) I feel there is a lack of explanation on how the framework, novelty of the method and theoretical results are an improvement of Tan, Wang, Liu and Zhang (2018, JRSSB).  In the latter paper, the same applications of SGEP are listed, the same Rayleigh quotient problem is formulated together with some bounds derived for the solution of SGEP. A more thorough discussion would be relevant. Moreover, these authors consider an optimisation that iteratively updates the solution based on the gradient of the generalised Rayleigh quotient and perform a truncation operation to foster sparsity. How does the proposed method clearly differ from Tan et al. (2018)?
2) What do the authors mean by "it holds asymptotically" in, e.g., page 4 line 133, or page 5 line 168? Does it mean that consistency holds (that is \tilde{A} converges in probability to A, p fixed?)?
3) page 7, the term "oracle properties" is used: this can be misleading since such expression can refer to asymptotic support recovery and asymptotic normal distribution of the non-zero parameters (following the definition of Fan and Li (2001)); or do the authors simply mean support recovery, that is identifying the true sparse support with high probability?
4) the notations are sometimes not clear: \lambda is used for the regularisation parameter and the eigenvalues of the true matrix B are denoted by \lambda_1, \lambda_2, ....
- On the theoretical results:
1) all the results are deterministic and the proofs are based mostly on optimisation reasoning. Would it be possible to evaluate the probability for which the bounds hold?
2) in both Theorems 1 and 2, the consistency highly depends on the behaviour of \eps, whose rate is a function of the error between the noisy measured matrix and the true matrix. As emphasised in "connection to the literature", the statistical viewpoint can provide some idea on the calibration of \eps, which governs the statistical accuracy and would provide some scaling among (n,p,cardinality(S)). It would be relevant to discuss how the consistency of \tilde{A} and \tilde{B} can be used to deduce the calibration of \lambda.
3) Theorem 4 is a key result for support recovery. To establish such a result, it is not clear whether the true parameter is assumed as the unique optimum of the non-penalised problem. The authors consider potentially non-convex penalty functions and potentially non-convex loss in their problem (10): there, the Hessian of the non-penalised loss x' \tilde{A} x can potentially be non-p.d. especially when n<p. What conditions ensure that the estimated optimum is close enough to the theoretical optimum and the existence of local/global optima?
Moreover, to deal with support recovery with non-convex regularisation, the Primal Dual Witness (PDW) method is usually performed to establish support recovery. This approach can not be empirically performed as it requires the knowledge of the true sparse subset. It seems in the proof of Theorem 4 that such sparse support is assumed known in the same vein as in PDW: a more detailed explanation of the proof steps would be more than welcome to support the theory and/or how these steps are similar to some related works on support recovery with non-convex penalisation.

**Time Spent Reviewing:**

Approximately 5 hours

---

> ### Author Response · Authors · 2021-08-10
> **Response to Reviewer bjrj**
>
> Thank you for your detailed comments. Given below are our point to point response.
>
>
> ---- On the clarity ----
>
> Q: Explanation on the framework, novelty of the method and theoretical results.
>
> A: Thanks for this question! This work aims to provide new insights to sgep problem from various perspectives.
> In the first part, we stand on basis of the perturbation analysis.
> We establish a new lower bound of the distance between perturbed solution and true solution.
> Such result fills the gap in the literature of perturbation analysis since most work only talk about the upper bound of
> perturbation error. No work gives the lower bound result in the sparse setting.
> In the second part, we stand on basis of estimation analysis.
> We try to understand the properties of sgep estimator via using non-convex penalties. To the best of our knowledge, there is
> no result on non-convex sgep yet. Unlike Tan, Wang, Liu and Zhang (2018, JRSSB) where they treat problem with hard constraints (truncation),
> our method is more like to give a soft support constraint and is less sensitive to the choice of sparse level.
>
> Q: What do the authors mean by "it holds asymptotically?
>
> A: Yes, it means that $\tilde{A}$ converges in probability to $A$, $p$ fixed.
>
> Q: The meaning of "oracle properties"?
>
> A: Sorry for confusion. It refers to support recovery, that is identifying the true sparse support with high probability.
>
> Q: Some notations are not clear.
>
> A: Sorry for unclear notations. We will use different notation for regularization parameter.
>
>
> ---- On the theoretical results ----
>
> Q: All the results are deterministic and the proofs are based mostly on optimisation reasoning.  Would it be possible to evaluate the probability for which the bounds hold?
>
> A: Thanks for raising this point. It is possible for evaluating the high probability bound via considering stochastic setting. We will add a section for this in the final version.
>
>
> Q: It would be relevant to discuss how the consistency of $\tilde{A}$ and $\tilde{B}$
>  can be used to deduce the calibration of $\lambda$.
>
> A: Thanks for this point. From statistical side, we require $\lambda >> \sqrt{s \log p / n}$ when the true Crawford number is bounded. We will add such result in the final version.
>
>
> Q:  On Theorem 4.
>
> A: The true parameter is not assumed to be a unique optimum of the non-penalized problem. But the true
> parameter is assumed to be unique in the sense of Identifiability condition (B4). In other words,
> the true parameter is the unique optimum point with support size less than $s_n$. In our perspective,
> (B4) is a reasonable condition even when $n << p$. Thanks for pointing out the primal dual witness (PDW) method. On the theoretical side, we prove the estimator is consistent by showing no other sparse candidate with different support can lead to smaller objective value compared with true parameter. This share the similar spirit as the PDW method. We agree that we do not know the true support empirically, therefore we only provide a local convergence result in Theorem 6.

---

### Official Review · Reviewer_4aA2 · 2021-07-16

**Rating:** 7
**Confidence:** 3

**Summary:**

The paper studies a class of sparse generalized eigenproblems which encompasses several common problems in machine learning. For this class of problems it gives upper and lower perturbation bounds which are shown to be rate-optimal. The authors then present an optimization problem with a non-convex penalty function and show that it can be used to estimate the sparse leading eigenvector. The theoretical results are validated numerically using synthetic data in the experimental section.




**Limitations And Societal Impact:**

The paper discusses some limitations in the final section. As the results of the paper are of methodological and theoretical nature, there is no direct societal impact.

**Main Review:**

Originality:

The main contribution of the paper is to give upper and lower bounds of the approximation error in the sparse generalized eigenproblem setting, given noisy observations. Moreover, it gives guarantees when the  sparsity structure of the true eigenvector can be recovered which is a novel result as well. Thus it fills some gaps in the literature.

Quality:

The paper seems to be technically sound and its main claims are well supported: In addition to the theoretical results, the derived perturbation bounds as well as the result of the recovery of the correct support are validated experimentally on synthetic data.

Clarity:

The manuscript is clearly written and well-organized.

Significance:

The class of problems studied in the paper contains many widely used techniques such as Sparse Principal Component Analysis, Sparse Fisher's Discriminant Analysis and Sparse Canonical Correlation Analysis. Therefore the paper covers a useful range of problems which might be of interest to the community. Moreover, the theoretical analysis of the sparse generalized eigenproblem given noisy observations fills some gaps in the literature and thus may lead to further work by other researchers.

----
After rebuttal:

I have read the feedback by the other reviewers and the responses by the authors. While the other reviewers pointed out some issues regarding the clarity of presentation, overall my impression of the paper is unchanged and I therefore kept my score.

**Time Spent Reviewing:**

4

---

> ### Author Response · Authors · 2021-08-10
> **Response to Reviewer 4aA2**
>
> Thank you for your positive feedback. We will continue to improve the writing of paper.

---

### Official Review · Reviewer_6VbX · 2021-07-30

**Rating:** 6
**Confidence:** 3

**Summary:**

In this paper, the authors consider the sparse generalized eigenvalue problem. The main contibutions of this paper are to establish the optimal rate of the error bound for estimating the sparse leading eigenvector and its support.


**Limitations And Societal Impact:**

Please refer to the points listed above in Main Review.

**Main Review:**

This paper focuses on the the sparse generalized eigenvalue problem. This problem is interesting. The main contributions of this paper are from the theoretical part. However, this paper is not well-writting, and there are several concerns listed below to be solved on the theoretical results of this paper.

1. There is lack of the proof of Theorem 2. Though the authors claimed that the proof of Theorem 2 can follow immediately from the proof of Lemma 2, we can observe that Lemma 2 presents the lower bound over the parameter space $F_\epsilon$, while the lower bound of Theorem 2 is obtained over $F_\epsilon, l$, which incorporates the sparsity structure and is much more complicated than $F_\epsilon$.

2. The authors do not make clear an important point: what is the fundamental difference of the optimal rates between the general setting and sparse setting discussed in Section 3.1 and 3.2. It is expected to incorporate the sparsity parameter in Theorem 2 for the lower bound under the sparse setting.

3. The authors claimed to obtain the optimal rate because the upper bound obtained in Theorem 1 matches the lower bound obtained in Theorem 2. However, the upper and lower bounds do not exactly match with each other, because the latter is obtained through two special cases.





**Time Spent Reviewing:**

7

---

> ### Author Response · Authors · 2021-08-10
> **Response to Reviewer 6VbX**
>
> Thank you for your comments. Given below are our point to point responses. Hope our response can address your concerns about our technical soundness.
>
>
> Q: Lack of proof of Theorem 2.
>
> A: If $\delta=0$, by using the case $\mathcal L = \emptyset$ in the proof of Theorem 1, then case (a) of Theorem 2 follows immediately from Lemma 2.
> If $\epsilon=0$,  $\tilde{w}_1$ in the proof of Theorem 1 is $u_1$.
> Using the proof for the upper bound  (A23),  we can show case (b) of Theorem 2 — simply use Lemma 2 for lower bound instead of Lemma 1 for upper bound.
> We will include this explanation in the final version.
>
>
> Q:  What is the fundamental difference of the optimal rates between the general setting and sparse setting?
>
> A: The fundamental difference between the general setting and sparse setting is discussed in the remark on line 165.
> In the general setting, the bound only has one term, which is at the order of the perturbation level;
> In the sparse setting, the upper bound has two terms, one is due to the perturbation; the other is due to the sparsity (failure in recovery of the true support set).
> In Theorem 2, the sparsity parameters are  indeed incorporated in the lower bound.
> Recall the definition of $A_{\mathcal I}$ on line 76, the condition number $\kappa_2(B_{\mathcal K})$ and the Crawford number  $c(A_{\mathcal K}, B_{\mathcal K})$ are both functions of $\mathcal{K}$.
>
> Q: The upper and lower bounds do not exactly match with each other, because the latter is obtained through two special cases?
>
> A: Thanks for asking this question.
>     There are two factors that affect the upper and lower bounds — the perturbation level and the failure in revealing the true support set of $u_1$, which are measured by $\epsilon$ and $\delta$, respectively.
> The upper bound (given by Theorem 1) is of the form $c_1 \epsilon + c_2 \delta$.
> If $\delta=0$, the upper bound matches the lower bound given by case (a) in Theorem 2;
> If $\epsilon=0$, the upper bound matches the lower bound given by case (b) in Theorem 2.
> Ideally, Theorem 2 should provide a lower bound of the form $c_3 \epsilon + c_4 \delta$ rather than lower bounds in two special cases.
> However, it is not an easy task. This is due to the triangular inequality:
> When establishing the upper bound, we show that  $|\sin\theta(u_1, x_*)| \le  |\sin\theta(u_1, w)| + |\sin\theta(w, x_*)| $ for some vector $w$.
> The first term  $|\sin\theta(u_1, w)|$ is at the order of $\epsilon$, the second term  $|\sin\theta(w, x_*)|$  is at the order of $\delta$.
> However,  for the lower bound, we only have $|\sin\theta(u_1, x_*)| \ge  | |\sin\theta(u_1, w)| - |\sin\theta(w, x_*)|  |$, which is not of the ideal form.
> Therefore, under the current framework of the proof, it is impossible show the ideal form.
> In addition, provided that one show the lower bound is of the form $f(\epsilon) + g(\delta)$ for certain functions f and g,
> taking $\delta=0$ and $\epsilon=0$, Theorem 2 tells that $f(\epsilon) = O(\epsilon)$ and $g(\delta)=O(\delta)$.
> Thus, the bound we established in Theorem 1 is still rate optimal.

---

> > ### Comment · Reviewer_6VbX · 2021-08-31
> > **Thanks for the reply**
> >
> > I have read the response, and increased the score to 6. But it is expected to obtain the lower bound in a unified case instead of two separate cases.

---

> ### Author Response · Authors · 2021-08-27
> **Additional Response to Reviewer 6VbX**
>
> Dear reviewer,  is there any further question regarding with technical proof of theoretical part. We are happy to provide more explanations. Thank you!

---

### Decision · Program_Chairs · 2021-09-27

**Decision:**

Accept (Poster)

**Comment:**

The main contribution of the paper is to give matching upper and lower bounds of the approximation error in the sparse generalized eigenproblem setting. This is an important contribution. To help better position the contribution it maybe best the authors add an explicit discussion on why the bounds are optimal.